# Turbocharging Solution Concepts: Solving NEs, CEs and CCEs with Neural Equilibrium Solvers

**Luke Marris**
DeepMind (`marris@deepmind.com`), UCL

| **Ian Gemp** | **Thomas Anthony** | **Andrea Tacchetti** | **Siqi Liu** | **Karl Tuyls** |
| --- | --- | --- | --- | --- |
| DeepMind | DeepMind | DeepMind | DeepMind, UCL | DeepMind |

## Abstract

Solution concepts such as Nash Equilibria, Correlated Equilibria, and Coarse Correlated Equilibria are useful components for many multiagent machine learning algorithms. Unfortunately, solving a normal-form game could take prohibitive or non-deterministic time to converge, and could fail. We introduce the Neural Equilibrium Solver which utilizes a special equivariant neural network architecture to approximately solve the space of all games of fixed shape, buying speed and determinism. We define a flexible equilibrium selection framework, that is capable of uniquely selecting an equilibrium that minimizes relative entropy, or maximizes welfare. The network is trained without needing to generate any supervised training data. We show remarkable zero-shot generalization to larger games. We argue that such a network is a powerful component for many possible multiagent algorithms.

## 1 Introduction

Normal-form solution concepts such as Nash Equilibrium (NE) [28, 46], Correlated Equilibrium (CE) [3], and Coarse Correlated Equilibrium (CCE) [45] are useful components and subroutines for many multiagent machine learning algorithms. For example, value-based reinforcement learning algorithms for solving Markov games, such as Nash Q-learning [27] and Correlated Q-Learning [19] maintain state action values for every player in the game. These action values are equivalent to per-state normal-form games, and policies are equilibrium solutions of these games. Critically, this policy will need to be recomputed each time the action-value is updated during training, and for large or continuous state-space Markov games, every time the agents need to take an action. Another class of multiagent algorithms are those in the space of Empirical Game Theoretic Analysis (EGTA) [60, 61] including PSRO [35, 44], JPSRO [40], and NeuPL [39, 38]. These algorithms are capable of training policies in extensive-form games, and require finding equilibria of empirically estimated normal-form games as a subroutine (the "meta-solver" step). In particular, these algorithms have been critical in driving agents to superhuman performance in Go [54], Chess [55], and StarCraft [59].

Unfortunately, solving for an equilibrium can be computationally complex. NEs are known to be PPAD [9, 8]. (C)CEs are defined by linear constraints, and if a linear objective is used to select an equilibrium, can be solved by linear programs (LPs) in polynomial time. However, in general, the solutions to LPs are non-unique (e.g. zero-sum games), and therefore are unsuitable equilibrium selection methods for many algorithms, and unsuitable for training neural networks which benefit from unambiguous targets. Objectives such as Maximum Gini [41] (a quadratic program (QP)), and Maximum Entropy [48] (a nonlinear program), are unique but are more complex to solve.

As a result, solving for equilibria often requires deploying iterative solvers, which theoretically can scale to large normal-form games but may (i) take an unpredictable amount of time to converge, (ii)

take a prohibitive amount of time to do so, and (iii) may fail unpredictably on ill-conditioned problems. Furthermore, classical methods [16, 36, 43] (i) do not scale, and (ii) are non-differentiable. This limits the applicability of equilibrium solution concepts in multiagent machine learning algorithms.

Therefore, there exists an important niche for approximately solving equilibria in medium sized normal-form games, quickly, in batches, reliably, and in a deterministic amount of time. With appropriate care, this goal can be accomplished with a Neural Network which amortizes up-front training cost to map normal-form payoffs to equilibrium solution concepts quickly at test time. We propose the Neural Equilibirum Solver (NES). This network is trained to optimize a composite objective function that weights accuracy of the returned equilibrium against auxiliary objectives that a user may desire such as maximum entropy, maximum welfare, or minimum distance to some target distribution. We introduce several innovations into the design and training of NES so that it is efficient and accurate. Unlike most supervised deep learning models, NES avoids the need to explicitly construct a labeled dataset of (game, equilibrium) pairs. Instead we derive a loss function that can be minimized in an unsupervised fashion from only game inputs. We also exploit the duality of the equilibrium problem. Instead of solving for equilibria in the primal space, we train NES to solve for them in the dual space, which has a much smaller representation. We utilize a training distribution that efficiently represents the space of all normal-form games of a desired shape and use an invariant preprocessing step to map games at test time to this space. In terms of the network architecture, we design a series of layers that are equivariant to symmetries in games such as permutations of players and strategies, which reduces the number of training steps and improves generalization performance. The network architecture is independent of the number of strategies in the game and we show interesting zero-shot generalization to larger games. This network can either be pretrained before being deployed, trained online alongside another machine learning algorithm, or a mixture of both.

## 2  Preliminaries

**Game Theory**   Game theory is the study of the interactive behaviour of rational payoff maximizing agents in the presence of other agents. The environment that the agents operate in is called a game. We focus on a particular type of single-shot, simultaneous move game called a *normal-form* game. A normal-form game consists of $N$ players, a set of strategies available to each player, $a_p \in \mathcal{A}_p$, and a payoff for each player under a particular joint action, $G_p(a)$, where $a = (a_1, ..., a_N) = (a_p, a_{-p}) \in \mathcal{A} = \otimes_p \mathcal{A}_p$. The subscript notation $-p$ is used to mean "all players apart from player $p$". Games are sometimes referred to by their shape, for example: $|A_1| \times ... \times |A_N|$. The distribution of play is described by a joint $\sigma(a)$. The goal of each player is to maximize their expected payoff, $\sum_{a \in \mathcal{A}} \sigma(a) G_p(a)$. Players could play independently by selecting a strategy according to their marginal $\sigma(a_p)$ over joint strategies, such that $\sigma(a) = \otimes_p \sigma(a_p)$. However this is limiting because it does not allow players to coordinate. A mediator called a *correlation device* could be employed to allow players to execute arbitrary joint strategies $\sigma(a)$ that do not necessarily factorize into their marginals. Such a mediator would sample from a publicly known joint $\sigma(a)$ and secretly communicate to each player their recommended strategy. Game theory is most developed in a subset of games: those with two players and a restriction on the payoffs, $G_1(a_1, a_2) = -G_2(a_1, a_2)$, known as zero-sum. Particularly in N-player, general-sum games, it is difficult to define a single criterion to find solutions to games. One approach is instead to consider joints that are in equilibrium: distributions such that no player has incentive to unilaterally deviate from a recommendation.

**Equilibrium Solution Concepts**   Correlated Equilibria (CEs) [3] can be defined in terms of linear inequality constraints. The deviation gain of a player is the change in payoff the player achieves when deviating to action $a'_p$ from a recommended action $a''_p$, when the other players play $a_{-p}$.

$$A_p^{\mathrm{CE}}(a'_p, a''_p, a) = A_p^{\mathrm{CE}}(a'_p, a''_p, a_p, a_{-p}) = \begin{cases} G_p(a'_p, a_{-p}) - G_p(a''_p, a_{-p}) & a_p = a''_p \\ 0 & \text{otherwise} \end{cases} \quad (1)$$

A distribution, $\sigma(a)$, is in $\epsilon$-CE if the deviation gain is no more than some constant $\epsilon_p \leq \epsilon$ for every pair of recommendation, $a''_p$, and deviation strategies, $a'_p$, for every player, $p$. These linear constraints geometrically form a convex polytope of feasible solutions.

$$\epsilon\text{-CE:} \qquad \sum_{a \in \mathcal{A}} \sigma(a) A_p^{\mathrm{CE}}(a'_p, a''_p, a) \leq \epsilon_p \qquad \forall p \in [1, N], a''_p \neq a'_p \in \mathcal{A}_p \qquad (2)$$

Coarse Correlated Equilibria (CCEs) [45] are similar to CEs but a player may only consider deviating before receiving a recommended strategy. Therefore the deviation gain for CCEs can be derived from the CE definition by summing over all possible recommended strategies $a_p''$.

$$A_p^{\text{CCE}}(a_p', a) = \sum_{a_p'' \in \mathcal{A}_p} A_p^{\text{CE}}(a_p', a_p'', a) = G_p(a_p', a_{-p}) - G_p(a) \qquad (3)$$

A distribution, $\sigma(a)$, is in $\epsilon$-CCE if the deviation gain is no more than some constant $\epsilon_p \leq \epsilon$ for every deviation strategy, $a_p'$, and for every player, $p$.

$$\epsilon\text{-CCE:} \qquad \sum_{a \in \mathcal{A}} \sigma(a) A_p^{\text{CCE}}(a_p', a) \leq \epsilon_p \qquad \forall p \in [1, N], a_p' \in \mathcal{A}_p \qquad (4)$$

NEs [46] have similar definitions to CCEs but have an extra constraint: the joint distribution factorizes $\otimes_p \sigma(a_p) = \sigma(a)$, resulting in nonlinear constraints[1].

$$\epsilon\text{-NE:} \qquad \sum_{a \in \mathcal{A}} \otimes_q \sigma(a_q) A_p^{\text{CCE}}(a_p', a) \leq \epsilon_p \qquad \forall p \in [1, N], a_p' \in \mathcal{A}_p \qquad (5)$$

Note that the definition of the NE uses the same deviation gain as the CCE definition. Another remarkable fact is that the marginals of any joint CCE in two-player constant-sum games, $\sigma(a_p) = \sum_{a_{-p}} \sigma(a)$, is also an NE, when $\epsilon_p = 0$. Therefore we can use CCE machinery to solve for NEs in such classes of games.

When a distribution is in equilibrium, no player has incentive to *unilaterally* deviate from it to achieve a better payoff. There can however be many equilibria in a game, choosing amongst these is known as the *equilibrium selection problem* [20]. For (C)CEs, the valid solution space is convex (Figure 1), so any strictly convex function will suffice (in particular, Maximum Entropy (ME) [48] and Maximum Gini (MG) [40] have been proposed). For NEs, the solution space is convex for only certain classes of games such as two-player constant-sum games. Indeed, NEs are considered fundamental in this class where they have powerful properties, such as being unexploitable, interchangeable, and tractable to compute. However, for N-player general-sum games (C)CEs may be more suitable as they remain tractable and permit coordination between players which results in higher-welfare equilibria. If $\epsilon_p \geq 0$, there must exist at least one NE, CE, and CCE for any finite game, because a NE always exists and NE $\subseteq$ CE $\subseteq$ CCE. Learning NEs [44] and CCEs [21] on a single game is well studied.

**Neural Network Solvers** Approximating NEs using neural networks is known to be PAC learnable [14]. There is also work learning (C)CEs [4, 32] and training neural networks to approximate NEs [14, 22] on subclasses of games. Learned NE meta-solvers have been deployed in PSRO [17]. Differentiable neural networks have been developed to learn QREs [37]. NEs for contextual games have been learned using fixed point (deep equilibrium) networks [24]. A related field, L2O [7], aims to learn an iterative optimizer more suited to a particular distribution of inputs, while this work focuses on learning a direct mapping. To the best of our knowledge, no work exists training a general approximate mapping from the full space of games to (C)CEs with flexible selection criteria.

## 3 Maximum Welfare Minimum Relative Entropy (C)CEs

Previous work has argued that having a unique objective to solve for equilibrium selection is important. The principle of maximum entropy [30] has been used to find unique equilibria [48]. In maximum entropy selection, payoffs are ignored and selection is based on minimizing the distance to the uniform distribution. This has two interesting properties: (i) it makes defining unique solutions easy, (ii) the solution is invariant transformations (such as offset and positive scaling) of the payoff tensor. While these solutions are unique, they both result in weak and low payoff equilibria because they find solutions on the boundary of the polytope. Meanwhile, the literature tends to favour Maximum Welfare (MW) because it results in high value for the agents and is a linear objective, however in general it is not unique. We consider a composite objective function composed of (i) Minimum Relative Entropy (MRE, also known as Kullback-Leibler divergence) between a target joint, $\hat{\sigma}(a)$,

---

[1]This is why NEs are harder to compute than (C)CEs.

and the equilibrium joint, $\sigma(a)$, (ii) distance between a target approximation, $\hat{\epsilon}_p$, and the equilibrium approximation, $\epsilon_p$, (iii) maximum of a linear objective, $W(a)$. The objective is constrained by the (i) distribution constraints ($\sum_a \sigma(a) = 1$ and $\sigma(a) \geq 0$) and, (ii) either CCE constraints (Equation (4)) or CE constraints (Equation (2)).

$$\arg\max_{\sigma,\epsilon_p} \mu \sum_{a \in \mathcal{A}} \sigma(a) W(a) - \sum_{a \in \mathcal{A}} \sigma(a) \ln\left(\frac{\sigma(a)}{\hat{\sigma}(a)}\right) - \rho \sum_p \left(\epsilon_p^+ - \epsilon_p\right) \ln\left(\frac{1}{\exp(1)} \frac{\epsilon_p^+ - \epsilon_p}{(\epsilon_p^+ - \hat{\epsilon}_p)}\right) \tag{6}$$

The approximation weight, $\rho$, and welfare weight, $\mu$, are hyperparameters that control the balance of the optimization. The maximum approximation parameter, $\epsilon_p^+$, is another constant that is usually chosen to be equal to the payoff scale (Section 4.1). The approximation term is designed to have a similar form to the relative entropy and is maximum when $\hat{\epsilon}_p = \epsilon_p$. We refer to this equilibrium selection framework as Target Approximate Maximum Welfare Minimum Relative Entropy ($\hat{\epsilon}$-MWMRE).

### 3.1 Dual of $\epsilon$-MWMRE (C)CEs

Rather than performing a constrained optimization, it is easier to solve the dual problem, $\arg\min_{\alpha_p} L^{\text{(C)CE}}$ (derived in Section A), where $\alpha_p^{\text{CE}}(a_p', a_p'') \geq 0$ are the dual deviation gains corresponding to the CE constraints, and $\alpha_p^{\text{CCE}}(a_p') \geq 0$ are the dual deviation gains corresponding to the CCE constraints. Note that we do not need to optimize over the primal joint, $\sigma(a)$. Choosing one of the elements in the curly brackets, the Lagrangian is defined:

$$\overset{\text{(C)CE}}{L} = \ln\left(\sum_{a \in \mathcal{A}} \hat{\sigma}(a) \exp\left(\overset{\text{(C)CE}}{l(a)}\right)\right) + \sum_p \epsilon_p^+ \left\{\sum_{a_p',a_p''} \overset{\text{CE}}{\alpha_p}(a_p', a_p''), \sum_{a_p'} \overset{\text{CCE}}{\alpha_p}(a_p')\right\} - \rho \sum_p \overset{\text{(C)CE}}{\epsilon_p} \tag{7}$$

The logits $l^{\text{(C)CE}}(a)$ are defined:

$$\overset{\text{(C)CE}}{l(a)} = \mu \sum_a W(a) - \sum_p \left\{\sum_{a_p',a_p} \overset{\text{CE}}{\alpha_p}(a_p', a_p) \overset{\text{CE}}{A_p}(a_p', a_p'', a), \sum_{a_p'} \overset{\text{CCE}}{\alpha_p}(a_p') \overset{\text{CCE}}{A_p}(a_p', a)\right\} \tag{8}$$

The primal joint and primal approximation parameters are defined:

$$\overset{\text{(C)CE}}{\sigma(a)} = \frac{\hat{\sigma}(a) \exp\left(\overset{\text{(C)CE}}{l(a)}\right)}{\sum_{a \in \mathcal{A}} \hat{\sigma}(a) \exp\left(\overset{\text{(C)CE}}{l(a)}\right)} \quad (9) \qquad \overset{\text{(C)CE}}{\epsilon_p} = (\hat{\epsilon}_p - \epsilon_p^+) \exp\left(-\frac{1}{\rho}\left\{\sum_{a_p',a_p''} \overset{\text{CE}}{\alpha_p}(a_p', a_p''), \sum_{a_p'} \overset{\text{CCE}}{\alpha_p}(a_p')\right\}\right) + \epsilon_p^+ \tag{10}$$

## 4 Neural Network Training

The network maps the payoffs of a game, $G_p(a)$, and the targets ($\hat{\sigma}(a)$, $\hat{\epsilon}_p$, $W(a)$) to the dual deviation gains, $\alpha_p^{\text{(C)CE}}$, that define the equilibrium. The duals are a significantly more space efficient objective target ($\sum_p |\mathcal{A}_p|^2$ for CEs and $\sum_p |\mathcal{A}_p|$ for CCEs) than the full joint ($\prod_p |\mathcal{A}_p|$), particularly when scaling the number of strategies and players. The joint, $\sigma(a)$, and approximation, $\epsilon_p$, can be computed analytically from the dual deviation gains and the inputs using Equations (9) and (10). The network is trained by minimizing the loss, $L^{\text{(C)CE}}$ (Equation (7)). We call the resulting architecture a Neural Equilibrium Solver (NES).

### 4.1 Input and Preprocessing

The MRE objective, and the (C)CEs constraints are invariant to payoff offset and scaling. Therefore we can assume that the payoffs have been standardized without loss of generality. Each player's payoff should have zero-mean. Furthermore, it should be scaled such that the $L_m = || \cdot ||_m$ norm of the payoff tensor equals $Z_m$, where $Z_m$ is a scale hyperparameter chosen such that the elements of

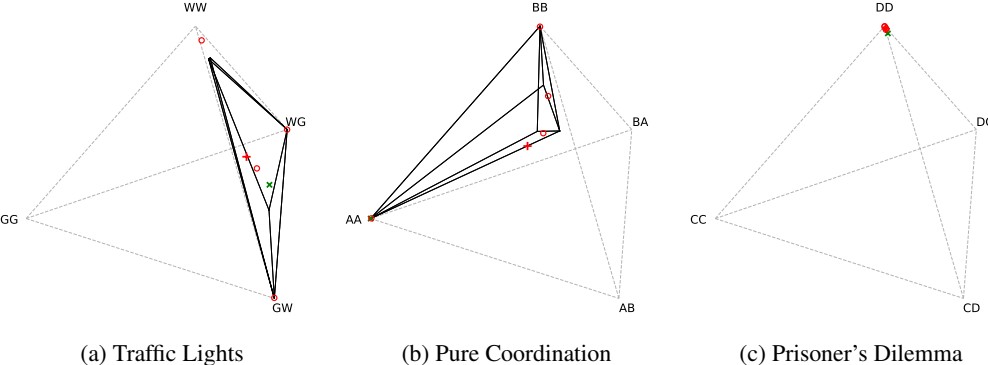

(a) Traffic Lights      (b) Pure Coordination      (c) Prisoner's Dilemma

Figure 1: Diagrams for three $2 \times 2$ normal-form games, showing their (C)CE solution polytope on the joint simplex (in two-strategy games CEs and CCEs are equivalent). An MWME NES, trained by sampling over the space of payoffs and welfare targets, is used to approximate the MW(C)CE solution ($\times$). An MRE NES, trained by sampling over the space of payoffs and joint targets, is used to approximate the ME(C)CE ($+$), and all pure joint target MRE(C)CEs ($\circ$). The networks have never trained on these games.

the inputs have unit variance (a property that ensures neural networks train quickly with standard parameter initializations [18]). We will ensure both these properties by including a preprocessing layer in NES. The preprocessed inputs ($G_p(a)$, $\hat{\sigma}(a)$, $\hat{\epsilon}_p$, $W(a)$) are then broadcast and concatenated together so that they result in an input of shape $[C, N, |\mathcal{A}_1|, ..., |\mathcal{A}_N|]$, where the channel dimension $C = 4$, if all inputs are required.

## 4.2 Training Distribution

The literature favours sampling games from the uniform or normal distribution. This introduces two problems: (i) it biases the distribution of games solvable by the network, and (ii) unnecessarily requires the network to learn offset and scale invariance in the payoffs. Recall that the space of equilibria are invariant to offset and positive scale transformations to the payoff. Zero-mean and $L_2$ norm scaling geometrically results in the surface of an $(|\mathcal{A}| - 1)$-ball centered on the origin (Section D.4). We propose using this invariant subspace for training. We choose the norm scaling constant, $Z_2 = \sqrt{|\mathcal{A}|}$, such that the elements of the payoffs maintain unit variance.

## 4.3 Gradient Calculation

The gradient update is found by taking the derivative of the loss (Equation (7)) with respect to the dual variables, $\alpha$. Note that computing a gradient does not require knowing the optimal joint $\sigma^*(a)$, so the network can be trained in an unsupervised fashion, from randomly generated inputs, $G_p(a)$, $\hat{\sigma}(a)$, $\hat{\epsilon}_p$, and $W(a)$.

$$\frac{\partial L^{\text{(C)CE}}}{\partial \left\{ \alpha_p^{\text{CE}}(a'_p, a''_p), \alpha_p^{\text{CCE}}(a'_p) \right\}} = \overset{\text{(C)CE}}{\epsilon_p} - \sum_a \left\{ \overset{\text{CE}}{A_p}(a'_p, a''_p, a), \overset{\text{CCE}}{A_p}(a'_p, a) \right\} \overset{\text{(C)CE}}{\sigma(a)} \quad (11)$$

The dual variables, $\left\{ \alpha_p^{\text{CE}}(a'_p, a''_p), \alpha_p^{\text{CCE}}(a'_p) \right\}$, are outputs of the neural network, with learned parameters $\theta$. Gradients for these parameters can be derived using the chain rule:

$$\frac{\partial L^{\text{(C)CE}}}{\partial \theta} = \frac{\partial L^{\text{(C)CE}}}{\partial \left\{ \alpha_p^{\text{CE}}(a'_p, a''_p), \alpha_p^{\text{CCE}}(a'_p) \right\}} \frac{\partial \left\{ \alpha_p^{\text{CE}}(a'_p, a''_p), \alpha_p^{\text{CCE}}(a'_p) \right\}}{\partial \theta}$$

Backprop efficiently calculates these gradients, and many powerful neural network optimizers [57, 34, 15] and ML frameworks [1, 5, 49] can be leveraged to update the network parameters.

## 4.4 Equivariant Architectures

The ordering of strategies and players in a normal-form game is unimportant, therefore the output of the network should be equivariant under two types of permutation; (i) strategy permutation, and (ii)

player permutation. Specifically, for some strategy permutation $\tau_p(1), ..., \tau_p(|\mathcal{A}_p|)$ applied to each element of a player's inputs (payoffs, target joint, and welfare), the outputs must also have permuted dimensions: $(\alpha_p(a_p) = \alpha_p(\tau_p(a_p))$ and $\sigma^\tau(a_1, ..., a_N) = \sigma(\tau_1(a_1), ..., \tau_N(a_N)))$. Likewise, for some player permutation $\tau(1), ..., \tau(N)$, the outputs must be transposed: $(\alpha_p^\tau(a_p) = \alpha_{\tau(p)}(a_{\tau(p)})$ and $\sigma^\tau(a_1, ..., a_N) = \sigma(a_{\tau(1)}, ..., a_{\tau(N)}))$. The latter equivariance can only be exploited by a network if all players have the same number of strategies ("cubic games"). There are $|\mathcal{A}_p|!$ possible strategy permutations for each player and $N!$ player permutations, resulting in $N! \left(|\mathcal{A}_p|!\right)^N$ possible equivariant permutations of each sampled payoff. Note that this is much greater than the number of joint strategies in a game, $N! \left(|\mathcal{A}_p|!\right)^N \gg |\mathcal{A}_p|^N$, which is an encouraging observation when considering how this approach will scale to large games. Utilizing an equivariant [52, 62] architecture is therefore crucial to scale to large games because each sample represents many possible inputs. Equivariant architectures have been used before for two-player games [22].

**Payoffs Transformations**   The main layers of the architecture consist of activations with shape $[C, N, |\mathcal{A}_1|, ..., |\mathcal{A}_N|]$, which is the same shape as a payoff tensor (with a channel dimension). We refer to layers with this shape as "payoff" layers. We consider transformations of the form:

$$g_{l+1}(c_{l+1}, p, a_1, ..., a_N) = f \left( \sum_{c_l^i}^{IC_l} w(c_{l+1}, c_l^i) \text{Con}_i^I \left[ \phi_i \left( g_l(c_l, p, a_1, ..., a_N) \right) \right] + b(c_{l+1}) \right) \quad (12)$$

where $f$ is any equivariant nonlinearity[2], $w$ are learned network weights, $b$ are learned network biases, and $\phi_i$ is one of many possible equivariant pooling functions (Section C) and Con is the concatenate function along the channel dimension. For example, consider one such function, $\phi_i = \sum_{a_1}$, which is invariant across any permutation of $a_1$ (similar to sum-pooling in CNNs), and equivariant over permutations of $p, a_2, ..., a_N$. In general we can use $\phi_{\subseteq \{p, a_1, ... a_N\}} g(p, a_1, ..., a_N)$ (mean-pooling and max-pooling are good choices). If all players have an equal number of strategies, for some functions, weights can be shared over all $p \in [1, N]$ because of symmetry [53]. Note that the number of trainable parameters scales with the number of input and output channels, and not with the size of the game (Figure 2), therefore it is possible for the network to generalize to games with different numbers of strategies. The basic layer, $g_{l+1}$, therefore comprises of a linear transform of a concatenated, broadcasted set of pooling functions.

**Payoffs to CCE Duals Transformations**   Payoffs can be transformed to CCE duals, $\alpha_p^{\text{CCE}}(c_{l+1}, a_p')$, by using a combination of a subset of the equivariant functions $\phi_i$ discussed above that sum over at least $-p$. If the number of strategies are equal for each player, the transformation weights can be shared and the duals can be stacked into a single object for more efficient computation in later layers: $\alpha^{\text{CCE}}(c_{l+1}, p, a_p') = \text{Stack}_p \left( \alpha_p^{\text{CCE}}(c_{l+1}, a_p') \right)$.

**Payoffs to CE Duals Transformations**   The transformation to produce the CE duals is more complex. CE duals, $\alpha_p^{\text{CCE}}(c_{l+1}, a_p'', a_p')$, need to be *symmetrically equivariant*. This property can be obtained by (i) independently generating two CCE duals and, (ii) taking outer operations, $\boxdot$ (for example sum or product), over them.

$$\alpha_p^{\text{CE}}(c_{l+1}, a_p'', a_p') = \hat{f} \left( \alpha_p^{\text{CCE}}(c_{l+1}, a_p'') \boxdot \alpha_p^{\text{CCE}}(c_{l+1}, a_p') \right) \quad (13)$$

Where $\hat{f}$ is any equivariant nonlinearity *with zero diagonal*[3]. We know that the diagonal is zero because it represents the dual of the deviation gain when deviating from a strategy to itself, which is zero, and therefore cannot be violated. This is a useful property which will be exploited in later dual layers. These can also be stacked if players have equal number of strategies: $\alpha^{\text{CCE}}(c_{l+1}, p, a_p', a_p'') = \text{Stack}_p \left( \alpha_p^{\text{CCE}}(c_{l+1}, a_p', a_p'') \right)$.

**CCE Duals Transformations**   Because the payoff activations are high-dimensional, it is worthwhile to operate on them in dual space. When transforming CCE duals we consider a mapping:

$$\alpha_{l+1}^{\text{CCE}}(c_{l+1}, p, a_p') = f \left( \sum_{c_l^i}^{IC_l} w(c_{l+1}, c_l^i) \text{Con}_i^I \left[ \phi_i \left( \alpha_l^{\text{CCE}}(c_l, p, a_p') \right) \right] + b(c_{l+1}) \right) \quad (14)$$

---

[2]Common nonlinearities such as element-wise (ReLU, tanh, sigmoid), and SoftMax are all equivariant.

[3]Masking is sufficient: e.g. $\hat{f}(a_p', a_p'') = (1 - I(a_p', a_p'')) f(a_p', a_p'')$, where $I$ is the identity matrix.

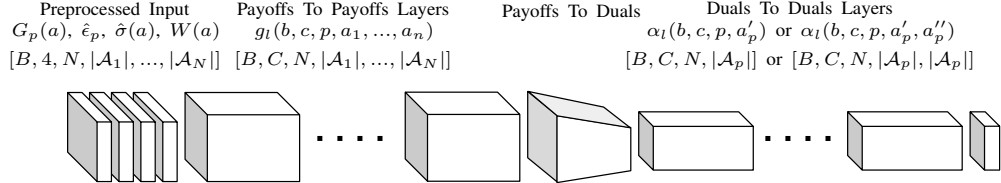

Preprocessed Input
$G_p(a)$, $\hat{\epsilon}_p$, $\hat{\sigma}(a)$, $W(a)$
$[B, 4, N, |\mathcal{A}_1|, ..., |\mathcal{A}_N|]$

Payoffs To Payoffs Layers
$g_l(b, c, p, a_1, ..., a_n)$
$[B, C, N, |\mathcal{A}_1|, ..., |\mathcal{A}_N|]$

Payoffs To Duals

Duals To Duals Layers
$\alpha_l(b, c, p, a'_p)$ or $\alpha_l(b, c, p, a'_p, a''_p)$
$[B, C, N, |\mathcal{A}_p|]$ or $[B, C, N, |\mathcal{A}_p|, |\mathcal{A}_p|]$

Figure 2: Network architecture showing the name, indices and shape (**B**atch, **C**hannels, **N**umber of players, **A**ctions per player) of each layer layer. Other architectures are possible, for example some of the inputs (target approximation, target joint, or welfare) could be passed in at a later layer.

where $f$ is any equivariant nonlinearity, and $\phi_i$ is from a set (Section C.2) of only two possible equivariant transformation functions (and two more if the game is cubic). For the final layer, we use a SoftPlus nonlinearity to ensure the output is nonnegative and has gradient everywhere.

**CE Duals Transformations**    When transforming CE duals we consider functions of the form:

$$\alpha^{\text{CE}}_{l+1}(c_{l+1}, p, a'_p, a''_p) = \hat{f}\left(\sum_{c_l}^{IC_l} w(c_{l+1}, c^i_l)\text{Con}^I_i\left[\phi_i\left(\alpha^{\text{CE}}_l(c_l, p, a'_p, a''_p)\right] + b(c_{l+l})\right)\right) \qquad (15)$$

For the CE case, the equivariant linear transformations are more complex: it is symmetric over the recommended and deviation strategies. Fortunately this is a well studied equivariance class [56], which can be fully covered by combining seven transforms (Section C) which comprise of different sums and transpositions of the input.

**Activation Variance**    Because the equivariant network possibly involves summing over dimensions of the inputs, activations are no longer independent of one another, so extra care needs to be taken when initializing the network to avoid variance explosion. To combat this we use three techniques: (i) inputs are scaled to unit variance as described previously (Section 4.1), (ii) the network is randomly initialized with variance scaling to ensure the variance at every layer is one, and (iii) we use BatchNorm [29] between every layer. We also used weight decay to regularize the network.

**Advanced Architectures**    More advanced architectures such as ResNet [23] or Transformers [58] are possible. An example of the final architecture is summarized in Figure 2.

### 4.5  Parameterizations

The composite objective framework allows us to define a number of combinations of auxiliary objectives. We highlight several interesting specifications (Appendix Table 3). The most basic is Maximum Entropy (ME) which simply finds the unique equilibrium closest to the uniform distribution according to the relative entropy distance. This distribution need not be uniform, it could be any target distribution. We could instead favour a welfare objective parameterized on the payoffs to find a Maximum Welfare (MW) solution. The two previous solutions can be generalized to solve for any welfare (Maximum Welfare Maximum Entropy (MWME)) or any target (Minimum Relative Entropy (MRE)). Furthermore we need not limit ourselves to approximation parameters equal to zero, for example by finding the minimum possible approximation parameter we have the Maximum Strength (MS) solution. Finally, we can find solutions for any approximation parameter for the solutions discussed so far ($\hat{\epsilon}$-MWME and $\hat{\epsilon}$-MRE).

## 5  Performance Experiments

Traditionally performance of NE, and (C)CE solvers has focused on evaluating time to converge to a solution within some tolerance. Feedforward neural networks can produce *batches* of solutions quickly[4] and deterministically. For non-trivial games this is much faster than what an iterative solver could hope to achieve. We therefore focus our evaluation on the trade-offs of the neural network

---

[4]We found inference, $\frac{\text{step time}}{\text{batch size}}$, to be around $1\mu s$ on our hardware.

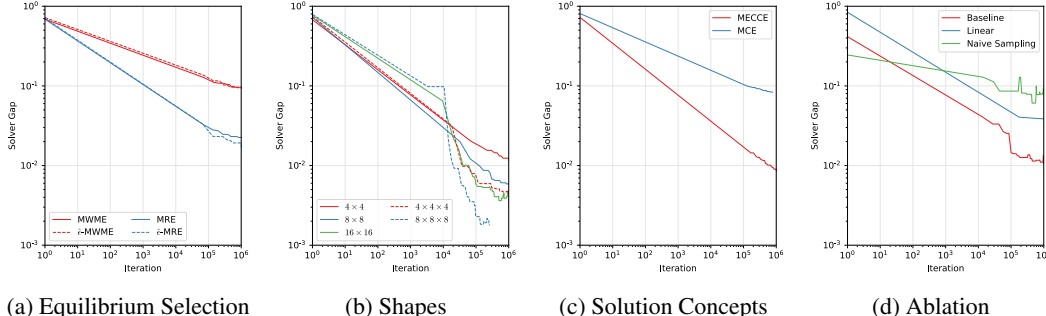

| (a) Equilibrium Selection | (b) Shapes | (c) Solution Concepts | (d) Ablation |

Figure 3: Sweeps and ablation studies showing the average solver gap of three experiment seeds evaluated over 512 sampled games against the number of train steps. Subfigure (a) shows $4 \times 4$ games over different equilibrium selection, (b) shows MECCE over games with different numbers of players and strategies, (c) shows CE and CCE concepts on $8 \times 8$ games, and (d) shows ablation experiments on MECCE $4 \times 4 \times 4$ games.

solver, namely (i) how long it takes to train, and (ii) how accurate the solutions are. For the latter we use two metrics:

$$\text{Solver Gap: } \frac{1}{2} \sum_a |\sigma^*(a) - \sigma(a)| \qquad \text{(C)CE Gap: } \sum_p \left[ \max_{\cdot} \sum_a (A_p(., a) - \epsilon_p) \right]^+ \sigma(a)$$

The first (Solver Gap) measures the distance to the exact unique solution found by an iterative solver[5], $\sigma^*(a)$, and is bounded between 0 and 1, and is zero for perfect prediction. The second ((C)CE Gap) measures the distance to the equilibrium solution polytope, and is zero if it is within the polytope.

**Parameterization Sweeps**   We show performance across a number of parameterizations, including (i) different equilibrium selection criteria (Figure 3a), (ii) different shapes of games (Figure 3b), and (iii) different solution concepts (Figure 3c).

**Classes of Games**   It is known that some distributions of game payoffs are harder for some methods to solve than others [51]. We compare performance across a number of classes of transfer games (Appendix Table 4) for a single MECCE and a single MECE [48] Neural Equilibrium Solver trained on $8 \times 8$ games. Figure 4 shows the worst, mean, and best performance over 512 samples from each class in terms of (i) distance to any equilibrium, and (ii) distance to the target equilibrium found by an iterative solver. We also plot the performance of a uniform joint as a naive baseline, as the gap can be artificially reduced by scaling the payoffs. In regards to equilibrium violation, ME is tricky because it lies on the boundary of the polytope, so some violation is expected in an approximate setting. The plots showing the failure rate and run time of the iterative solver are to intuit difficult classes. The baseline iterative solver take about $0.05$s to solve a single game, the network can solve a batch of 4096 games in $0.0025$s. We see that for most classes the NES is very accurate with a solver gap of around $10^{-2}$. Some classes of games are indeed more difficult and these align with games that iterative equilibrium solvers struggle with. This hints that difficult games are ill-conditioned.

**Ablations**   We show the performance (Figure 3d) of the proposed method compared with (i) a linear network, and (ii) no invariant pre-processing with naive payoff sampling (each element sampled using a uniform distribution). Both result in significant reduction in performance.

**Scaling**   Due to the size of the representation of the payoffs, $G_p(a)$, the inputs and therefore the activations of the network grow significantly with the number of joint strategies in the game. Therefore without further work on sparser payoff representation, NES is limited by size of payoff inputs. For further discussion see Section 7. Nevertheless, Table 1 shows good performance when scaling to moderately sized games. Note that the "solver gap" metric is incomplete on larger games because the ECOS evaluation solver fails to converge.

---

[5]We use an implementation in CVXPY [12, 2] which leverages the ECOS [13] solver.

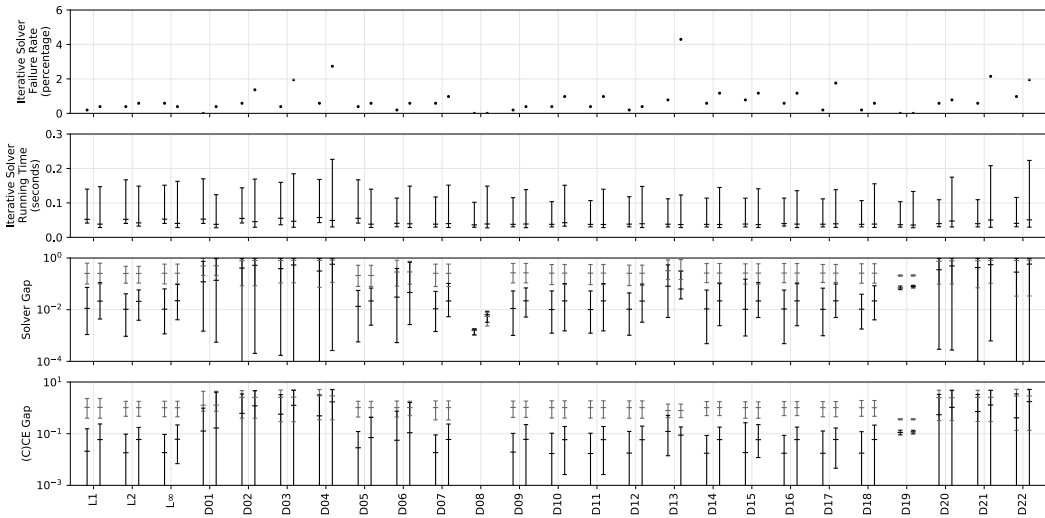

Figure 4: Worst, mean, and best performance of MECCE (left in pair) and MECE (right in pair) over 512 samples on the three classes introduced in this paper (Section 4.1), and on a subset [50] of transfer GAMUT [47] games (Appendix Table 4). The network was only trained on the "$L_2$ invariant subspace" distribution of games. The gray range indicates the performance under a uniform distribution baseline.

Table 1: Scaling experiments showing the gaps of five NES models for larger games, with a uniform baseline, over 128 samples. The ECOS solver used to evaluate "solver gap" fails on large games.

| Game | CCE Gap under uniform | CCE Gap under NES | Solver Gap under uniform | Solver Gap under NES | Success Fraction |
|---|---|---|---|---|---|
| $4 \times 4$ | 1.1006 | 0.0274 | 0.3552 | 0.0120 | 100% |
| $8 \times 8$ | 1.0043 | 0.0163 | 0.2513 | 0.0054 | 99% |
| $16 \times 16$ | 0.8861 | 0.0173 | 0.2014 | 0.0034 | 98% |
| $32 \times 32$ | 0.7376 | 0.0215 | – | – | 0% |
| $64 \times 64$ | 0.5864 | 0.0288 | – | – | 0% |

**Generalization**    An interesting property of the NES architecture is that its parameters do not depend on the number of strategies in the game. Therefore we can test the generalization ability of the network zero-shot on games with different numbers of strategies (Table 2). There are two observations: (i) NES only weakly generalizes to other game sizes under the solver gap metric, and (ii) NES strongly generalizes to larger games under the CCE gap, remarkably achieving zero violation. Therefore the network retains the ability to reliably find CCEs in larger games, but does struggle to accurately select the target MWMRE equilibrium. This could be mitigated by training the network on a mixture of game sizes, which we leave to future work.

## 6    Applications

With Neural Equilibrium Solvers (NES) it is possible to quickly find approximate equilibrium solutions using a variety of selection criteria. This allows the application of solution concepts into areas that would otherwise be too time-expensive and are not as sensitive to approximations.

**Inner Loop of MARL Algorithms**    For algorithms [28, 27, 19, 35, 41, 39] where speed is critical, and the size of games is modest, but numerous, and approximations can be tolerated.

**Warm-Start Iterative Solvers**    Many iterative solvers start with a guess of the parameters and refine them over time to find an accurate solution [14]. It is possible to use NES to warm-start iterative solver algorithms, potentially significantly improving convergence.

Table 2: Generalization experiments showing how an $8 \times 8$ network generalizes to games with a different number of strategies, over 128 samples.

| Game | CCE Gap under uniform | CCE Gap under NES | Solver Gap under uniform | Solver Gap under NES | Success Fraction |
|---|---|---|---|---|---|
| $4 \times 4$ | 1.1006 | 4.4445 | 0.3552 | 0.1500 | 100% |
| $\mathbf{8 \times 8}$ | 1.0043 | 0.0163 | 0.2513 | 0.0054 | 99% |
| $16 \times 16$ | 0.8861 | 0.0000 | 0.2014 | 0.1089 | 98% |
| $32 \times 32$ | 0.7376 | 0.0000 | – | – | 0% |
| $64 \times 64$ | 0.5864 | 0.0000 | – | – | 0% |

**Polytope Approximation**  The framework can be used to approximate the full space of solutions by finding extreme points of the convex polytope. Because of convexity, any convex mixture of these extreme points is also a valid solution. Two approaches could be used to find extreme points (i) using different welfare objectives (ii) or using different target joint objectives. For example, using pure joint targets:

$$W(a) = \begin{cases} 1, & \text{if } a = \hat{a} \\ 0, & \text{otherwise} \end{cases} \qquad \hat{\sigma}(a) = \begin{cases} 1^-, & \text{if } a = \hat{a} \\ 0^+, & \text{otherwise} \end{cases}$$

These could be computed in a single batch, and would cover a reasonably large subset of full polytope of solutions (the latter approach is demonstrated in Figure 1). It would be easy to develop an algorithm that refines the targets at each step to gradually find all vertices of the polytope, if desired.

**Differentiable Model and Mechanism Design**  Mechanism design (MD) is a sub-field of economics often described as "inverse game theory", where instead of studying the behavior of rational payoff maximizing agents on a given game, we are tasked with designing a game so that rational payoff maximizing participants will exhibit behaviours *at equilibrium* that we deem desirable. The field has a long history to which it is near impossible to do justice; see [42] for a review. The work presented here could impact MD in two ways. First, by making it easy to compute equilibrium strategies, NES could widen the class of acceptable output games, relaxing the restrictive requirements (e.g. strategic dominance) often imposed of the output games out of concern more permissive solution concepts could be hard for participants to compute. Second, NES maps payoffs to joint strategies and it is differentiable, one could imagine turning a mechanism design task to a optimization problem that could be solved using standard gradient descent (e.g. design a general-sum game where strategies at equilibrium maximize some non-linear function of welfare and entropy, with payoff lying in a useful convex and closed subset). A related idea is to find a game that produces a certain equilibrium [37]. Given an equilibrium, a payoff could be trained through the differentiable model that results in the desired specific behaviour.

## 7  Discussion

The main limitation of the approach is that the activation space of the network is large, particularly with a large number of players and strategies which limits the size of games that can be tackled. Future work could look at restricted classes of games, such as polymatrix games [10, 11], or graphical games [31], which consider only local payoff structure and have much smaller payoff representations. This is a promising direction because NES otherwise has good scaling properties: (i) the dual variables are space efficient, (ii) there are relatively few parameters, (iii) the number of parameters is independent of the number of strategies in the game, (iv) equivariance means each training sample is equivalent to training under all payoff permutations, and (v) there are promising zero-shot generalization results to larger games.

Solving for equilibria has the potential to promote increased cooperation in general-sum games, which could increase the welfare of all players. However, if a powerful and unethical actor had influence on the game being played, welfare gains of some equilibria could unfairly come at the expense of other players.

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
