# A Approximate Target Maximum Welfare Minimum Relative Entropy Equilbiria

We use a Minimum Relative Entropy (RME) (also known as minimum KL divergence) $\sum_a \sigma(a) \ln\left(\frac{\sigma(a)}{\hat{\sigma}(a)}\right)$, where $\hat{\sigma}(a) > 0$ is a full-support joint such that, $\sum_a \hat{\sigma}(a) = 1$. This objective is similar to Maximum Entropy Correlated Equilibrium (MECE) [48], and the proofs here are similar to the framework set out there. A drawback of MECE is that it is not easy to determine the minimum $\epsilon_p$ permissible. If we choose $\epsilon_p$ that does not permit a valid solution, then the parameters will diverge. We can circumvent this problem by optimizing the distance to a target $\hat{\epsilon}_p$. We engineer this target, $\min_{\epsilon_p} \rho \sum_p \left(\epsilon_p^+ - \epsilon_p\right) \ln\left(\frac{1}{\exp(1)} \frac{\epsilon_p^+ - \epsilon_p}{(\epsilon_p^+ - \hat{\epsilon}_p)}\right)$, to have a global minimum at $\epsilon_p = \hat{\epsilon}_p$, where $0 < \rho < \infty$ is a hyper-parameter used to control the balance between the distance to the target distribution and the distance to the target approximation parameter. And $\mu$ is for balancing the linear objective.

## A.1 CEs

**Theorem 1** ($\epsilon$-MWMRE CE). *The $\hat{\epsilon}$-MWMRE CE solution is equivalent to minimizing the loss:*

$$\overset{CE}{L} = \ln\left(\sum_{a\in\mathcal{A}} \hat{\sigma}(a) \exp\left(\overset{CE}{l}(a)\right)\right) + \sum_p \epsilon_p^+ \sum_{a_p', a_p''} \overset{CE}{\alpha_p}(a_p', a_p'') - \rho \sum_p \overset{CE}{\epsilon_p}$$

*With logits defined as:*

$$\overset{CE}{l}(a) = \mu \sum_a W(a) - \sum_{p,a_p',a_p''} \overset{CE}{\alpha_p}(a_p', a_p'') \overset{CE}{A_p}(a_p', a_p'', a)$$

*And primal variables defined:*

$$\overset{CE}{\sigma}(a) = \frac{\hat{\sigma}(a) \exp\left(\overset{CE}{l}(a)\right)}{\sum_{a\in\mathcal{A}} \hat{\sigma}(a) \exp\left(\overset{CE}{l}(a)\right)} \qquad \overset{CE}{\epsilon_p} = (\hat{\epsilon}_p - \epsilon_p^+) \exp\left(-\frac{1}{\rho} \sum_{a_p', a_p''} \overset{CE}{\alpha_p}(a_p', a_p'')\right) + \epsilon_p^+$$

*Proof.* Construct a Lagrangian, $\max_{\sigma,\epsilon} \min_{\alpha,\beta,\lambda} L_{\alpha_p,\beta,\lambda}^{\sigma,\epsilon_p}$, where the primal variables are being maximized and the dual variables are being minimized.

$$L_{\alpha_p,\beta,\lambda}^{\sigma,\epsilon_p} = -\sum_a \sigma(a) \ln\left(\frac{\sigma(a)}{\hat{\sigma}(a)}\right) + \mu \sum_{a\in\mathcal{A}} W(a)\sigma(a) - \rho \sum_p \left(\epsilon_p^+ - \epsilon_p\right) \ln\left(\frac{1}{\exp(1)} \frac{\epsilon_p^+ - \epsilon_p}{(\epsilon_p^+ - \hat{\epsilon}_p)}\right)$$

$$+ \sum_a \beta(a)\sigma(a) - \lambda\left(\sum_a \sigma(a) - 1\right) - \sum_{p,a_p',a_p''} \alpha_p(a_p', a_p)\left(\sum_a \sigma(a) A_p(a_p', a_p'', a) - \epsilon_p\right)$$

$$= \sum_a \sigma(a)\left(-\ln\left(\frac{\sigma(a)}{\hat{\sigma}(a)}\right) + \mu W(a) + \beta(a) - \lambda - \sum_{p,a_p',a_p''} \alpha_p(a_p', a_p'') A_p(a_p', a_p'', a)\right)$$

$$+ \sum_{p,a_p',a_p''} \alpha_p(a_p', a_p'')\epsilon_p + \lambda - \rho \sum_p \left(\epsilon_p^+ - \epsilon_p\right) \ln\left(\frac{1}{\exp(1)} \frac{\epsilon_p^+ - \epsilon_p}{(\epsilon_p^+ - \hat{\epsilon}_p)}\right)$$

Taking the derivatives with respect to the joint distribution $\sigma(a)$, and setting to zero.

$$\frac{\partial L_{\alpha_p,\beta,\lambda}^{\sigma,\epsilon_p}}{\partial \sigma(a)} = -\ln\left(\frac{\sigma(a)}{\hat{\sigma}(a)}\right) - 1 + \mu W(a) + \beta(a) - \lambda - \sum_{p,a_p',a_p} \alpha_p(a_p', a_p) A_p(a_p', a_p, a) = 0$$

$$\sigma^*(a) = \hat{\sigma}(a) \exp\left(-\lambda - 1 + \mu W(a) + \beta(a) - \sum_{p,a_p',a_p} \alpha_p(a_p', a_p) A_p(a_p', a_p, a)\right)$$

Substituting back in:

$$L_{\alpha_p,\beta,\lambda}^{\epsilon_p} = \sum_a \sigma^*(a) + \sum_{p,a'_p,a_p} \alpha_p(a'_p, a_p)\epsilon_p + \lambda - \rho \sum_p \left(\epsilon_p^+ - \epsilon_p\right) \ln\left(\frac{1}{\exp(1)} \frac{\epsilon_p^+ - \epsilon_p}{(\epsilon_p^+ - \hat{\epsilon}_p)}\right)$$

Taking the derivative with respect to $\lambda$ and setting to zero.

$$\frac{\partial L_{\alpha_p,\beta,\lambda}^{\epsilon_p}}{\partial \lambda} = -\sum_a \sigma^*(a) + 1 = 0$$

$$\exp\left(\lambda^* + 1\right) = \sum_a \hat{\sigma}(a) \exp\left(\mu W(a) + \beta(a) - \sum_{p,a'_p,a''_p} \alpha_p(a'_p, a''_p) A_p(a'_p, a''_p, a)\right)$$

Substituting back in:

$$L_{\alpha_p,\beta}^{\epsilon_p} = \ln\left(\sum_a \hat{\sigma}(a) \exp\left(\mu W(a) + \beta(a) - \sum_{p,a'_p,a''_p} \alpha_p(a'_p, a''_p) A_p(a'_p, a''_p, a)\right)\right)$$

$$+ \sum_{p,a'_p,a''_p} \alpha_p(a'_p, a''_p)\epsilon_p - \rho \sum_p \left(\epsilon_p^+ - \epsilon_p\right) \ln\left(\frac{1}{\exp(1)} \frac{\epsilon_p^+ - \epsilon_p}{(\epsilon_p^+ - \hat{\epsilon}_p)}\right)$$

Noting that the term is minimized when $\beta(a) = 0$, and that we can lift the $\hat{\sigma}(a)$ term into the exponential, we have:

$$L_{\alpha_p}^{\epsilon_p} = \ln\left(\sum_a \hat{\sigma}(a) \exp\left(\mu W(a) - \sum_{p,a'_p,a''_p} \alpha_p(a'_p, a''_p) A_p(a'_p, a''_p, a)\right)\right)$$

$$+ \sum_{p,a'_p,a''_p} \alpha_p(a'_p, a''_p)\epsilon_p - \rho \sum_p \left(\epsilon_p^+ - \epsilon_p\right) \ln\left(\frac{1}{\exp(1)} \frac{\epsilon_p^+ - \epsilon_p}{(\epsilon_p^+ - \hat{\epsilon}_p)}\right)$$

Taking the derivatives with respect to the approximation parameter $\epsilon_p$, and setting to zero.

$$\frac{\partial L_{\alpha_p}^{\epsilon_p}}{\partial \epsilon_p} = \rho \ln\left(\frac{\epsilon_p^+ - \epsilon_p^*}{\epsilon_p^+ - \hat{\epsilon}_p}\right) + \sum_{a'_p,a_p} \alpha_p(a'_p, a_p) = 0 \implies \epsilon_p^* = (\hat{\epsilon}_p - \epsilon_p^+) \exp\left(-\frac{1}{\rho} \sum_{a'_p,a_p} \alpha_p(a'_p, a_p)\right) + \epsilon_p^+$$

Therefore:

$$\ln\left(\frac{1}{\exp(1)} \frac{\epsilon_p^+ - \epsilon_p^*}{(\epsilon_p^+ - \hat{\epsilon}_p)}\right) = \ln\left(\frac{1}{\exp(1)} \exp\left(-\frac{1}{\rho} \sum_{a'_p,a_p} \alpha_p(a'_p, a''_p)\right)\right) = -\frac{1}{\rho} \sum_{a'_p,a''_p} \alpha_p(a'_p, a''_p) - 1$$

Substituting back in:

$$L_{\alpha_p} = \ln\left(\sum_a \hat{\sigma}(a) \exp\left(\mu W(a) - \sum_{p,a'_p,a''_p} \alpha_p(a'_p, a''_p) A_p(a'_p, a''_p, a)\right)\right)$$

$$+ \sum_{p,a'_p,a''_p} \epsilon_p^+ \alpha_p(a'_p, a''_p) - \rho \sum_p (\hat{\epsilon}_p - \epsilon_p^+) \exp\left(-\frac{1}{\rho} \sum_{a'_p,a''_p} \alpha_p(a'_p, a''_p)\right)$$

$\square$

## A.2 CCEs

**Theorem 2** (CCE). *The $\hat{\epsilon}$-MWMRE CCE solution is equivalent to minimizing the loss:*

$$\overset{CCE}{L} = \ln\left(\sum_{a \in \mathcal{A}} \hat{\sigma}(a) \exp\left(\overset{CCE}{l(a)}\right)\right) + \sum_p \epsilon_p^+ \sum_{a_p'} \overset{CCE}{\alpha_p}(a_p') - \rho \sum_p \overset{CCE}{\epsilon_p}$$

*With logits defined as:*

$$\overset{CCE}{l}(a) = \mu \sum_a W(a) - \sum_{p,a_p'} \overset{CCE}{\alpha_p}(a_p') \overset{CCE}{A_p}(a_p', a)$$

*And primal variables defined:*

$$\overset{CCE}{\sigma}(a) = \frac{\hat{\sigma}(a) \exp\left(\overset{CCE}{l(a)}\right)}{\sum_{a \in \mathcal{A}} \hat{\sigma}(a) \exp\left(\overset{CCE}{l(a)}\right)} \qquad \overset{CCE}{\epsilon_p} = (\hat{\epsilon}_p - \epsilon_p^+) \exp\left(-\frac{1}{\rho} \sum_{a_p'} \overset{CCE}{\alpha_p}(a_p')\right) + \epsilon_p^+$$

*Proof.* Similar proof to Theorem 1. □

# B  Unit Variance Scaling

The inputs are preprocessed so that the network does not need to learn to be invariant to the offset or scale. This is achieved by using a zero-mean offset, and normalizing by an $m$-norm, scaled with a constant $Z_m$. This constant is chosen such that the variance of the elements is one.

$$G_p^{L_m}(a) = Z_m \frac{G_p(a) - \frac{1}{|\mathcal{A}|} \sum_a G_p(a)}{\left\| G_p(a) - \frac{1}{|\mathcal{A}|} \sum_a G_p(a) \right\|_m} \tag{16a}$$

$$\hat{\epsilon}_p^{L_m} = \text{clip}\left(\frac{\hat{\epsilon}_p}{\left\| G_p(a) - \frac{1}{|\mathcal{A}|} \sum_a G_p(a) \right\|_m}, -\hat{\epsilon}^+ = -Z_m, +\hat{\epsilon}^+ = +Z_m\right) \tag{16b}$$

$$W^{L_m}(a) = Z_m \frac{W(a) - \frac{1}{|\mathcal{A}|} \sum_a W(a)}{\left\| W(a) - \frac{1}{|\mathcal{A}|} \sum_a W(a) \right\|_m} \tag{16c}$$

$$\hat{\sigma}^{L_1}(a) = Z_\sigma\left(\hat{\sigma}(a) - \frac{1}{|\mathcal{A}|}\right) \tag{16d}$$

Some scale factors are:

$$Z_\sigma = |\mathcal{A}| \sqrt{\frac{|\mathcal{A}| + 1}{|\mathcal{A}| - 1}} \qquad\qquad Z_2 = \sqrt{|\mathcal{A}|} \tag{17}$$

The constant, $Z_\sigma$, of the joint is derived from the variance of elements of a flat Dirichlet distribution. The constant, $Z_2$, of the the $L_2$ ring is derived from the mean of the Chi-Squared distribution.

# C  Equivariant Pooling Functions

Functions which maintain equivariance over player and strategy permutations are useful building blocks for neural network architectures. These comprise of two components; (i) an equivariant pooling function $\phi$, such as mean sum, min, or max, and (ii) the reduction dimensions.

An *equivairant pooling function* has three properties: (i) it collapses one or more of the dimensions of a tensor, (ii) the operation is invariant to the order of the elements in the collapsed dimensions, and

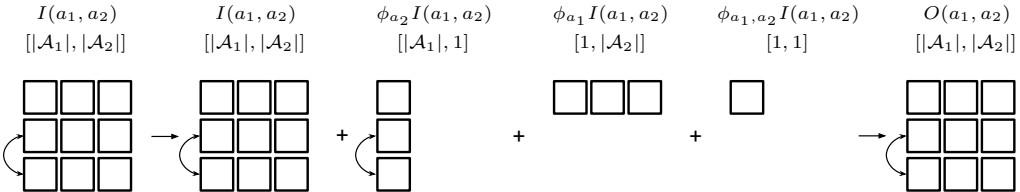

$$I(a_1, a_2) \qquad I(a_1, a_2) \qquad \phi_{a_2} I(a_1, a_2) \qquad \phi_{a_1} I(a_1, a_2) \qquad \phi_{a_1, a_2} I(a_1, a_2) \qquad O(a_1, a_2)$$
$$[|\mathcal{A}_1|, |\mathcal{A}_2|] \qquad [|\mathcal{A}_1|, |\mathcal{A}_2|] \qquad [|\mathcal{A}_1|, 1] \qquad [1, |\mathcal{A}_2|] \qquad [1, 1] \qquad [|\mathcal{A}_1|, |\mathcal{A}_2|]$$

Figure 5: Equivariant Pooling Functions, mapping an input $I(a_1, a_2)$ to an output $O(a_1, a_2)$. Swapping the second and third row (for example) of the input results in the same swap in the outputs.

(iii) the operation is equivariant to the order of the elements in the non-collapsed dimensions. For example, the reduction $\sum_{a_2} G_1(a_1, a_2) = R_1(a_1)$ (i) reduces the dimensionality from $|\mathcal{A}_1| \times |\mathcal{A}_2|$ to $|\mathcal{A}_1|$, (ii) reordering the columns of $G_1(a_1, a_2)$ does not change the calculation, but (iii) reordering the rows of $G_1(a_1, a_2)$ results in an equivariant output, where $R_1(a_1)$ is reordered in the same way.

Such combinations of pooling functions and reduction dimensions can be combined to construct a network that is equivariant to strategy and player permutation. Consider several such pooling functions composed together (Figure 5).

## C.1 Payoffs to Payoffs

Some equivariant functions which map payoff structures to payoff structures are:

$$g(p, a_1, ..., a_N) \quad (18a) \qquad \underset{p, a_q}{\phi}\, g(p, a_1, ..., a_N) \quad (18e) \qquad \underset{a_p}{\phi}\, g(q, a_1, ..., a_N) \quad (18i)$$

$$\underset{a_1, ..., a_N}{\phi}\, g(p, a_1, ..., a_N) \quad (18b) \qquad \underset{p, a_{-q}}{\phi}\, g(p, a_1, ..., a_N) \quad (18f) \qquad \underset{a_{-p}}{\phi}\, g(q, a_1, ..., a_N) \quad (18j)$$

$$\underset{p, a_1, ..., a_N}{\phi}\, g(p, a_1, ..., a_N) \quad (18c) \qquad \underset{a_q}{\phi}\, g(p, a_1, ..., a_N) \quad (18g) \qquad \underset{a_q}{\phi}\, g(q, a_1, ..., a_N) \quad (18k)$$

$$\underset{p}{\phi}\, g(p, a_1, ..., a_N) \quad (18d) \qquad \underset{a_{-q}}{\phi}\, g(p, a_1, ..., a_N) \quad (18h) \qquad \underset{a_{-q}}{\phi}\, g(q, a_1, ..., a_N) \quad (18l)$$

$$\forall\, q \in [1, N] \qquad\qquad\qquad \forall\, q \in [1, N]$$

If all players have an equal number of strategies, for some primitives (Equations (18e)-(18l)), weights can be shared over all $q \in [1, N]$ because of symmetry.

## C.2 CCE Duals to CCE Duals

All equivariant CCE dual pooling functions are given below. Equations (19c) and (19d) may only be used for cube shaped games.

$$\alpha_p(a'_p) \quad (19a) \qquad \underset{a'_p}{\phi}\, \alpha_p(a'_p) \quad (19b) \qquad \underset{p}{\phi}\, \alpha_p(a'_p) \quad (19c) \qquad \underset{p, a'_p}{\phi}\, \alpha_p(a'_p) \quad (19d)$$

## C.3 CE Duals to CE Duals

All zero-diagonal equivariant CE dual pooling functions are:

$$\alpha_p(a'_p, a''_p) \quad (20a) \qquad \underset{a'_p}{\phi}\, \alpha_p(a'_p, a''_p) \quad (20c) \qquad \underset{a''_p}{\phi}\, \alpha_p(a'_p, a''_p) \quad (20e) \qquad \underset{a'_p, a''_p}{\phi}\, \alpha_p(a'_p, a''_p) \quad (20g)$$

$$\alpha_p(a''_p, a'_p) \quad (20b) \qquad \underset{a'_p}{\phi}\, \alpha_p(a''_p, a'_p) \quad (20d) \qquad \underset{a''_p}{\phi}\, \alpha_p(a''_p, a'_p) \quad (20f) \qquad \underset{p, a'_p, a''_p}{\phi}\, \alpha_p(a'_p, a''_p) \quad (20h)$$

Table 3: Possible Neural Equilibrium Solver solution parameterizations.

| | $G_p(a)$ | $\hat{\sigma}(a)$ | $\hat{\epsilon}_p$ | $\hat{\epsilon}^+$ | $W(a)$ | $\rho$ | $\mu$ |
|---|---|---|---|---|---|---|---|
| ME | $\sim L_m$ | $\frac{1}{|\mathcal{A}|}$ | $0$ | $Z_m$ | $0$ | $\gg 1$ | $0$ |
| MT | $\sim L_m$ | $\hat{\sigma}(a)$ | $0$ | $Z_m$ | $0$ | $\gg 1$ | $0$ |
| MU | $\sim L_m$ | $\frac{1}{|\mathcal{A}|}$ | $0$ | $Z_m$ | $\sum_p G_p(a)$ | $\gg 1$ | $\gg 1$ |
| MWME | $\sim L_m$ | $\frac{1}{|\mathcal{A}|}$ | $0$ | $Z_m$ | $\sim L_m$ | $\gg 1$ | $\gg 1$ |
| MRE | $\sim L_m$ | $\sim \mathrm{Dir}(1)$ | $0$ | $Z_m$ | $0$ | $\gg 1$ | $0$ |
| MS | $\sim L_m$ | $\frac{1}{|\mathcal{A}|}$ | $-Z_m$ | $Z_m$ | $0$ | $\gg 1$ | $0$ |
| $\hat{\epsilon}_p$-ME | $\sim L_m$ | $\frac{1}{|\mathcal{A}|}$ | $\sim \mathrm{U}(-Z_m, Z_m)$ | $Z_m$ | $0$ | $\gg 1$ | $0$ |
| $\hat{\epsilon}_p$-MWME | $\sim L_m$ | $\frac{1}{|\mathcal{A}|}$ | $\sim \mathrm{U}(-Z_m, Z_m)$ | $Z_m$ | $\sim L_m$ | $\gg 1$ | $\gg 1$ |
| $\hat{\epsilon}_p$-MRE | $\sim L_m$ | $\sim \mathrm{Dir}(1)$ | $\sim \mathrm{U}(-Z_m, Z_m)$ | $Z_m$ | $0$ | $\gg 1$ | $0$ |

# D  Experiment Architecture and Hyper-Parameters

The architecture and hyper-parameters were chosen from a coarse sweep. The performance of architecture was not very sensitive to parameterization: similar settings will work well, or even better. Nevertheless we provide the details of the exact architecture used in the experiments.

## D.1  Architecture

All experiments use the same network architecture, with either CCE or CE dual parameterization, implemented in JAX [5] and Haiku [25]. We used pooling functions (Equations (18a)-(18d) and (18k)-(18l)) for the payoffs to payoffs layers, and used all the pooling functions for dual layers. For $\phi$ we used both mean and max pooling together. The we used 5 payoffs to payoffs layers, each with 32 channels, a payoffs to duals layer with 64 channels and 2 duals to duals layers with 32 channels, which we denote $[(32, 32, 32, 32, 32), 64, (32, 32)]$. The network has 79,905 parameters. All nonlinearities are ReLUs apart from the final layer where we use a Softplus. Between every layer we use BatchNorm [29] with learned scale and variance correction. The network was initialized such that the variance of activations at every layer is unity. This was done empirically by passing a dummy batch of data through the network and calculating the variance.

## D.2  Hyper-Parameters

We used a training batch size of 4096, the Optax [26] implementation of Adam [34] (learning rate $4 \times 10^{-4}$) optimizer with adaptive gradient clipping [6] (clipping $10^{-3}$). We used a learning rate schedule with (iteration, factor) pairs of $[(1 \times 10^5, 1.0), (1 \times 10^6, 0.6), (4 \times 10^6, 0.3), (7 \times 10^6, 0.1), (1 \times 10^7, 0.06), (1 \times 10^8, 0.03)]$. We included a weight decay loss (learning rate $1 \times 10^{-7}$).

## D.3  Network Parameterizations

Different possible network parameterizations can be found in Table 3.

## D.4  Game Distributions

A list of different game distributions can be found in Table 4. The geometric interpretation of the invariant subspace of games is shown in Figure 6.

## D.5  Hardware

We trained our network on a 32 core TPU v3 [33], and evaluated on an 8 core TPU v2 [33]. For intuition, the $8 \times 8$ network trains at around $400$ batches per second (1,638,400 examples per second). Evaluation is even faster. Bigger games take longer, and scales approximately linearly with the number of joint actions in the game.

Table 4: Classes of random games. We consider the three scale and offset invariant classes introduced in this paper (Section 4.1), and on a subset [50] of GAMUT [47] games using the functions in parenthesis and parameterized with the `-random_params` flag.

| Name | Game Description |
|---|---|
| $L_1$ | $L_1$ Invariant |
| $L_2$ | $L_2$ Invariant |
| $L_\infty$ | $L_\infty$ Invariant |
| D1 | Bertrand Oligopoly (`BertrandOligopoly`) |
| D2 | Bidirectional LEG, Complete Graph (`BidirectionalLEG-CG`) |
| D3 | Bidirectional LEG, Random Graph (`BidirectionalLEG-RG`) |
| D4 | Bidirectional LEG, Star Graph (`BidirectionalLEG-SG`) |
| D5 | Covariance Game, $\rho = 0.9$ (`CovariantGame-Pos`) |
| D6 | Covariance Game, $\rho \in [-1/(N-1), 1]$ (`CovariantGame`) |
| D7 | Covariance Game, $\rho = 0$ (`CovariantGame-Zero`) |
| D8 | Dispersion Game (`DispersionGame`) |
| D9 | Graphical Game, Random Graph (`GraphicalGame-RG`) |
| D10 | Graphical Game, Road Graph (`GraphicalGame-Road`) |
| D11 | Graphical Game, Star Graph (`GraphicalGame-SG`) |
| D12 | Graphical Game, Small-World (`GraphicalGame-SW`) |
| D13 | Minimum Effort Game (`MinimumEffortGame`) |
| D14 | Polymatrix Game, Complete Graph (`PolymatrixGame-CG`) |
| D15 | Polymatrix Game, Random Graph (`PolymatrixGame-RG`) |
| D16 | Polymatrix Game, Road Graph (`PolymatrixGame-Road`) |
| D17 | Polymatrix Game, Small-World (`PolymatrixGame-SW`) |
| D18 | Uniformly Random Game (`RandomGame`) |
| D19 | Travelers Dilemma (`TravelersDilemma`) |
| D20 | Uniform LEG, Complete Graph (`UniformLEG-CG`) |
| D21 | Uniform LEG, Random Graph (`UniformLEG-RG`) |
| D22 | Uniform LEG, Star Graph (`UniformLEG-SG`) |

Table 5: Payoffs for two games that show that maximum welfare cannot be discovered via a MW objective to the distribution given by a softmax of welfare. Both games are symmetric, payoffs are given for the row player

(a) CE MW Counterexample

|   | 1 | 2 | 3 | 4 |
|---|---|---|---|---|
| 1 | -4 | 2, -2 | -999 | -999 |
| 2 | -2, 2 | 1 | -999 | -999 |
| 3 | -999 | -999 | -3 | 2, -2 |
| 4 | -999 | -999 | -2, 2 | 1.1 |

(b) CCE MW Counterexample

|   | 1 | 2 | 3 | 4 |
|---|---|---|---|---|
| 1 | 2 | 0 | 0 | 0, 2 |
| 2 | 0 | 3 | 0, 3 | -10, 7 |
| 3 | 0 | 3, 0 | -10 | -10, -6 |
| 4 | 2, 0 | 7, -10 | -6, -10 | 0 |

# E  Relative Entropy and Welfare Objectives

Any solution can be realised by some relative entropy objective, since if a joint distribution $\sigma(a)$ is a (C)CE, then the solution with a relative entropy objective to $\hat{\sigma}(a) = \sigma(a)$ itself is optimised by $\sigma(a)$. One might imagine therefore that a relative entropy objective could be chosen to induce a maximum welfare solution, based on the payoffs. If possible, this would allow the MWMRE to be simplified. However, it is not straightforward to determine a priori which relative entropy objective(s) will lead to the maximum welfare. This means that relative entropy objectives are insufficient for finding Maximum Welfare solutions.

For example, consider a welfare $W(a) = \sum_p G_p(a)$. We might try to induce a welfare maximising solution by choosing a target joint $\hat{\sigma}(a) = \lim_{T \to \infty} \text{SoftMax}(TW(a))$, where $T$ is the temperature parameter. Finding a CE that minimizes relative entropy to $\hat{\sigma}$ would place high mass on the highest welfare joint action, but is not equivalent to maximizing the linear objective $\sum_a \sigma(a)W(a)$, for either CCEs or CEs.

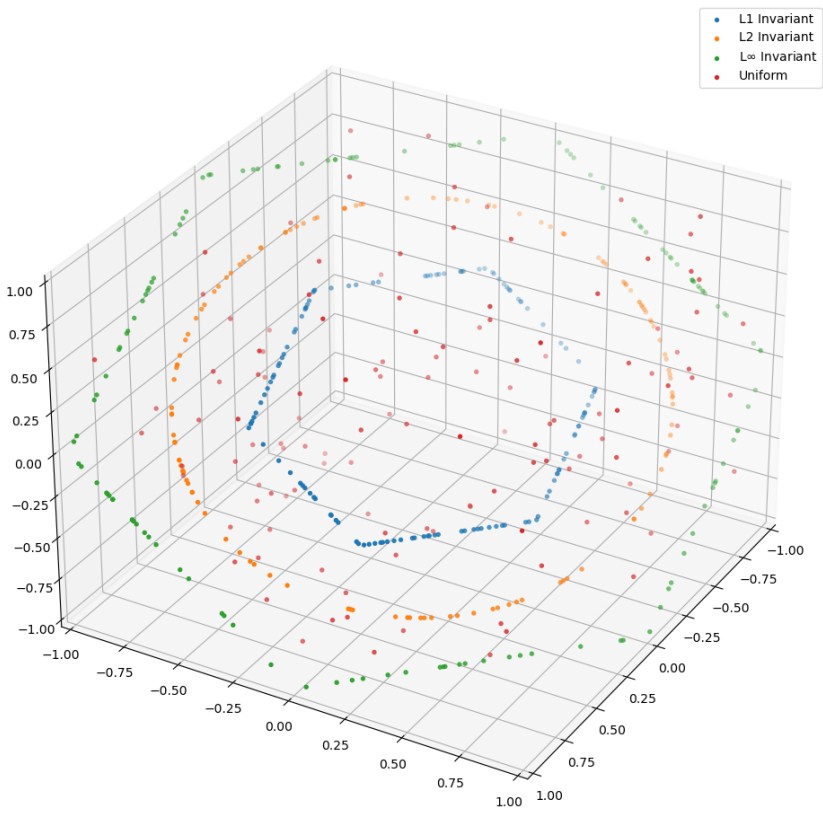

Figure 6: Shows 128 samples of payoffs for player 1 in $3 \times 1$ shaped games, under four different distributions. A $3 \times 1$ game is not theoretically interesting, but is used here because visualizing above 3 dimensions is difficult. Normalizing by an offset to result in zero-mean payoffs, and by a positive scale to result in unit norm payoffs, is geometrically the surface of an $(|\mathcal{A}| - 1)$-ball. It is straightforward to uniformly sample over such a space. Furthermore, no interesting game structure is lost by only considering this subspace, because offset and positive scale transformations are invariant transformations. It is easy to map any payoff to this invariant subspace, so the neural network can handle any game of appropriate shape at test time. Meanwhile, a naive sample method, such as uniformly sampling payoffs is an unwieldy input for a neural network to decode.

Consider game (a) in Table 5. This game consists of two games of chicken side-by-side, which are mutually incompatible (i.e. the players must co-ordinate to play the same game of chicken to avoid a very large negative payoff for both). The softmax relative entropy objective will prefer the action pair $(4, 4)$ over all others, as it gives slightly higher payoffs than $(2, 2)$. Notice that a CE that recommends $(4, 4)$ must also recommend the action pair $(4, 3)$ some of the time in order to disincentivise the row player from deviating from action 4 to action 3. Similarly, a CE that recommends $(2, 2)$ must also recommend $(2, 1)$ some of the time to disincentivise the row player from deviating from action 2 to action 1.

Crucially, because $(1, 1)$ has a lower payoff for the row player than $(3, 3)$, in CEs consisting $(2, 2)$ the mediator doesn't have to recommend $(2, 1)$ as often as it has to recommend $(4, 3)$ to form an effective disincentive. The result is that a CE that plays exclusively the joint actions $(1, 2)$, $(2, 1)$ and $(2, 2)$ can achieve higher welfare than any that plays $(4, 4)$, despite never playing the welfare maximising joint action.

Game (b) in Table 5 provides a counterexample for the CCE case. It works in a similar way to the CE counterexample: there are two high welfare joint strategies, $(1, 1)$ and $(2, 2)$. The latter has higher welfare, but if played too much a deviation to strategy 4 is incentivised. In the limit of $T$, the relative entropy objective selects whichever equilibrium has the highest probability of the max-welfare joint.

To disincentivse the row player's deviation to strategy 4, the column player must either: play only strategies 1 and 4, because the strategies 1 and 4 have the same payoff for the row player, or play strategy 3 sufficiently frequently that the benefit of deviating from strategy 2 to strategy 4 is nullified.

The first option gives rise to the max-welfare CCE, which plays (1, 1) with probability 1. The second gives rise to the CCE that plays (2, 2) with the highest possible probability: 0.2. It plays (2, 3) and (3, 2) with probability 0.4 each. This is chosen by the relative entropy objective, but gives each player an average payoff of 1.8, which is equal to the payoff for deviating to action 4 in this equilibrium, but lower than the payoff of (1, 1).