# OpenReview forum: "Turbocharging Solution Concepts: Solving NEs, CEs and CCEs with Neural Equilibrium Solvers"
_NeurIPS.cc/2022/Conference — NeurIPS 2022 Accept_

### Official Review · Reviewer_RFxv · 2022-07-10

**Rating:** 7
**Confidence:** 4
**Soundness:** 3 good
**Presentation:** 2 fair
**Contribution:** 3 good

**Summary:**

Games are a useful formalism in machine learning. There are multiple notions of solution to a game; this paper focuses on Correlated Equilibria (CE) and Coarse Correlated Equilibria (CCE). Given the payoffs of a game, the set of all CE (resp. CCE) form a polytope, with number of constraints exponential in the number of players and actions. Thus, solving for a CE (CCE) can be computationally intractable.

This paper proposes to use a neural network to predict approximate CE (resp. CCE) given the payoffs of all players, as well as some auxiliary data. The authors propose several innovations to make the proposed neural equilibrium solver more efficient and flexible:
  - several choices of secondary criterion used to select a particular CE (resp. CCE) from within the polytope of all CE (resp. CCE).
  - A permutation-invariant network architecture, capable of exploiting symmetries in the game (e.g. exchangeability of players, permutations of a players action set).
  - Focusing on solving the dual problem, which has fewer optimization variables as compared to the primary problem.



**Questions:**

Let me elaborate on some of the weaknesses highlighted above. If you can clarify my confusions, I will consider raising my score.

## Training and experiments.

  1. **Training Loss and gradients** I don't quite understand how you trained the proposed network. What is the loss function? Also, how do you propose computing gradients with respect to the network parameters? (Eq. 11 gives a formula for a gradient, yes, but it is not with respect to the parameters of the neural network.)

  2. **Training data** Can you tell me more about the training data set you used? As far as I can understand, you randomly generated a fresh i.i.d batch of 4096 $8\times 8$ games (I guess by randomly sampling payoff matrices) each iteration. Is this correct? Also, I found Section 4.2 a bit hard to follow, for example, what is meant by the first sentence "Zero-mean and constant norm scale geometrically results in the edge of a disc."?

3. **Train time and network size**  As you mention in Section 5, the train time is an important metric when evaluating learned solvers. Could you comment on how long (in wall-clock time) it took to train your NES for $8\times 8$ games? Also, how many parameters in total did this network have?

 3. **Scalability** I am a little skeptical that your approach will scale to large games. Indeed, your experiments seem to target small games ($4\times 4$, $8\times 8$). Could you run an experiment for truly large games, e.g. $100\times 100$ or $1000\times 1000$? If no, could you clarify the intended use-case of the proposed NES?

## Relation to Learn to Optimize literature

I appreciated the reduction of determining a CE/ CCE to a convex optimization problem. I also think the use of duality to reduce the number of variables is clever. However after this reduction, the problem you're proposing: learning a solver for an optimization problem is well-studied in the Learn-to-Optimize (L2O) literature. Thus, the relation of your work to this literature should be discussed (see, for example [1] and citations therein). Using some of the terminology in [1] may help to describe your training set-up and resolve my questions above.

## Relation to Learning to solve Games literature.

Several recent works [2,3,4] have proposed uses gradient-based learning in conjunction with game solvers. The focus of these papers is slightly different to yours---closer to the inverse game theory literature briefly alluded to in the conclusion. Nevertheless, the relation between your work and these papers should be discussed.

## Minor Comments
  1. Lines 19--21 are needlessly repetitive: "take a non-deterministic amount of time to converge", "take a prohibitive amount of time to do so", and "do not scale" are all essentially the same point.

  2. Citation [10] should be updated. It seems the paper has changed name. A related comment: the citation numbers in the main article and the supplementary do not match.

  3. In line 101, when introducing relative entropy, you should mention that this concept is also known as Kullback-Liebler divergence (you actually do this in the appendix).

  4. Line 106. Typo in the word "hyperparameter".

  5. Line 113--114 I think it is more precise to say you solve the dual problem instead of "optimize the dual Lagrangian".

  6. In line 131--132 you mention $Z_m$, but it's not clear what this is yet. You could clarify that this is just some user-defined scale parameter.

  7. Line 137. I found this sentence hard to understand, as mentioned above.

  8. Line 215. Is "Maximum Utilitarian" the same as "Maximum welfare"?

  9. What do the axes mean in Figure 3? Is the y-axis average solver gap over a test set of games? Is the x-axis number of training iterations?

  10. Line 252--253. "would be otherwise be too time-expensive" has an extra "be".

## Bibliography
[1] Chen *et al* Learning to Optimize: A Primer and A Benchmark (2021)

[2] Ling *et al* What game are we playing? End-to-end learning in normal and extensive form games (2018)

[3] Li *et al* End-to-End Learning and Intervention in Games (2020)

[3] Heaton *et al* Learn to Predict Equilibria via Fixed Point Networks (2021)



**Limitations:**

These are adequately addressed.

**Strengths And Weaknesses:**

## Strengths
  - Predicting/ computing NE/CE/CCE is certainly a worthy topic.
  - I appreciated the use of strong duality for convex optimization!
  - Experiments are thorough.

## Weaknesses
  - Overlooks certain important aspects of the literature.
  - I found it hard to follow how the proposed network is actually trained.
  - Scalability of proposed approach to even moderately sized games is unclear.

##Edit
Changed score to 7 after rebuttal.

---

> ### Author Response · Authors · 2022-08-02
> **Reviewer #3 Rebuttal**
>
> Thank you for recognizing the importance and elegance of the solution, for your constructive criticism and, in particular, your reference recommendations. We have worked to answer your questions and hope it has improved your opinion of our work and will be available during the discussion period to answer any follow-up questions.
>
> ### Gradients
>
> The loss function is given in Equation (7). This is parameterized by the dual variables, $\alpha$, which are the output of the neural network. Everything else in Equation (7) is either a user-defined hyper-parameter or an input to the neural network. This loss function alone is enough for frameworks like TensorFlow, Jax, or PyTorch to determine the gradients through auto-differentiation. Section 4.3 (Equation (11)) shows the first step of differentiating the loss with respect to the duals, $\frac{\partial L}{\partial \alpha}$. We have clarified in this section that the network parameters can be derived using the chain rule, $\frac{\partial L}{\partial \theta} = \frac{\partial L}{\partial \alpha}\frac{\partial \alpha}{\partial \theta}$, and efficiently computed using backprop. Please see the updated section, we hope this clarifies this issue.
>
> ### Training Distribution
>
> We argue that sampling payoffs naively (for example using a uniform or normal distribution) is harder for a neural network to learn (Figure 3(d)) because the neural network would (i) unnecessarily need to precisely learn about offset and scale invariances and (ii) the uniform and normal distributions impose an unnatural bias on the distribution of payoffs that the model can handle. We propose sampling from a zero-mean, constant-norm-scale space, which we refer to as “Disc” distributions. This space (i) contains all interesting games because offset and positive scale transformations do not change the equilibrium space of a game, (ii) It can be easily uniformly sampled from, (iii) any game can be easily mapped to this space before calculating its equilibrium at test time, and (iv) it imposes fewer biases on the training distribution (for example the network would not have worse predictions of payoffs with high scale just because they are not in the training distribution).
>
> To give further intuition about the geometry of the disc distributions we have added a visualization in Figure 5 (appendix Section D.4). Aside: we are considering renaming “L2 Disc” to “L2 N-1 Sphere”, because the term is generating confusion.
>
> You are correct. We randomly generate iid batches of games (payoff arrays of shape [NUM_PLAYERS, NUM_ACTIONS_PLAYER_1,...,NUM_ACTIONS_PLAYER_N]) from the L2 disc distribution (the batch size is a user chosen hyper-parameter, we used 4096 in our experiments), and train the network using standard gradient based optimizers. Because of the definition of our loss function, we do not need to generate target solutions, and can train in an unsupervised fashion.
>
> ### Wall Clock
>
> The 8x8 network trains at around 400 steps per second. Therefore it takes 40 minutes to train 1 million iterations. Bigger games take longer, and scales approximately linearly with the number of joint actions in the game.
>
> ### Network Parameters
>
> The network has 79,905 parameters.
>
> ### Scalability
>
> Please see our response to reviewer 2.
>
> ### L2O
>
> Our approach is not L2O because we are learning a direct mapping, not a distribution specific optimization algorithm. We have included a reference to this literature because we agree that it is related, and L2O could feasibly be applied to this problem.
>
> ### Learning to solve Games literature
>
> Reference [2] is very interesting and certainly relevant. Thank you for bringing it to our attention. [3] is also interesting. Both have been added to the literature review.
>
> ### Minor Comments
>
> Thank you for spotting these problems. We have incorporated these fixes.

---

> > ### Comment · Reviewer_RFxv · 2022-08-07
> > **Response to Rebuttal**
> >
> > Thank you for your comprehensive response to my concerns! Also, please accept my apologies for this late response to your rebuttal; I have been away from keyboard.
> >
> > ## Gradients
> > OK, I now understand. I appreciate section 4.3!
> >
> > ## Training Distribution
> > Thanks, I now understand this much better. Section D.4 adds a lot of clarity. I would recommend using the terminology "L2 N-1 Sphere" or "boundary of the L2 ball" as I think this would be clearer.
> >
> > ## Wall Clock time
> > Thanks! I would include this in the paper, as in my interpretation it falls under point 3(d) of the checklist ("include the total amount of compute")
> >
> > ## L2O and Learning to Solve Games
> > I reflected upon this and I agree with you; your work is not L2O as you are not learning an iterative solver.  I guess I would characterize your work as learning a really good initialization, when viewed from an L2O perspective (which you do mention in Section 6). Thus, your work may be of use to this community too.
> >
> > I think your discussion of the relation between your work and the L2O literature in Section 2 is brief but good.
> >
> > ## Scalability
> > This was my main hesitation in reviewing this work. I have read your discussion with Reviewer 2 and I think I have a clearer understanding of the motivation for, and potential use case of, your work. NES is designed for solving large batches of small games to moderate accuracy, not solving large games or solving to high accuracy. Would you say this is accurate? I am not an expert on MARL, but I can see that could be a compelling application of NES. Could you make this clearer in the introduction? (I see you have added a sentence on this to Section 7, but I think it would help readers appreciate the significance of your work if your also mentioned this at the beginning of the paper).
> >
> > Looking at your paper with fresh eyes I believe it is significant enough to appear at NeuRIPS. I think my initial score was too harsh, and I apologize for this. So,  **I am raising my score to accept**. Thank you again for your comprehensive rebuttal. I will check in again tomorrow, in case there is anything further to discuss.

---

> > > ### Author Response · Authors · 2022-08-08
> > > **Reply**
> > >
> > > No worries. Thank you for raising your score.
> > >
> > > >  I would recommend using the terminology "L2 N-1 Sphere" or "boundary of the L2 ball" as I think this would be clearer.
> > >
> > > We are  happy to do this for camera ready.
> > >
> > > > I would include this in the paper, as in my interpretation it falls under point 3(d) of the checklist ("include the total amount of compute")
> > >
> > > Likewise, we are happy to include this information in a camera ready version.
> > >
> > > > I guess I would characterize your work as learning a really good initialization
> > >
> > > > This was my main hesitation in reviewing this work. I have read your discussion with Reviewer 2 and I think I have a clearer understanding of the motivation for, and potential use case of, your work. NES is designed for solving large batches of small games to moderate accuracy, not solving large games or solving to high accuracy. Would you say this is accurate?
> > >
> > > Yes this is accurate. We aspire to train the network as accurately as possible - certainly accurate enough for a variety of applications - but not to the tolerances achieved by iterative solvers. The method is not intended for precisely solving large games. We will be careful to position the contribution as such - our story in this draft could have been clearer in this regard.
> > >
> > > > I am not an expert on MARL, but I can see that could be a compelling application of NES. Could you make this clearer in the introduction? (I see you have added a sentence on this to Section 7, but I think it would help readers appreciate the significance of your work if your also mentioned this at the beginning of the paper).
> > >
> > > Thank you for the suggestion. We are happy to include this as a motivating example.

---

### Official Review · Reviewer_duLd · 2022-07-11

**Rating:** 6
**Confidence:** 4
**Soundness:** 3 good
**Presentation:** 3 good
**Contribution:** 3 good

**Summary:**

This paper aims to create a method for training NNs to solve for NE, CE, and CCE across a set of games with a fixed action space for each player.  This is achieved through considering the dual formulation of the LP resulting from each of these equilibria concepts, and preforming gradient decent on the NN given random games in with that action space size so that the strategies the network produce satisfy the constraints of the LP.

**Questions:**

How were the hyperparmeters selected for your approach v.s the baselines?


**Limitations:**

The limitations were adequately addressed except to the extent mentioned above.

**Strengths And Weaknesses:**

**Strengths:**
The strengths of this paper are
* The novel (to my knowledge) idea to train solvers of games in the dual LP space rather than the primal.  This is an interesting approach, which could be promising and should be explored.
* The idea to train one solver for all games of a given shape rather than solving each game on-by-one
* Equivariant networks and payoff invariances which are shown to improve performance.

**Weaknesses:**
The main weaknesses I see are:
* The motivation that other approaches in the space cannot solve for CE seems weak since it seems easy to adapt PSRO or CFRM to such a class.
* The scaling argument seems weak as the approach of training on all games of a give size seems much much more difficult computationally than solving a particular game in that class.
* If the argument is that this method is scalable, then I would expect experiments that run on much larger than 8 by 8 games.

More broadly, as this paper is taking the approach of solving all N by N games at once, but most of the payoff matrices in this space are ones which we don't use in practice, I would expect an evaluation on a "transfer set" of games which we do care about to show that it works to solve them correctly.  Otherwise it is difficult to evaluate the method, as it could have "low error on average" but not low error on the problems we care about.

In the same vein, since we usually care about getting the CE for a particular game, I would be useful to compare to finding the CE for that particular game directly rather than solving all games of that shape first.

**Minor Comments:**
Figure 4 is hard to understand, given that the  difference between the gray and black error bars are never described.

The appeal to "Occam's razor" on line 93 is very suspect.  It appears this is alluding to something specific (which should be cited), but regardless of what the citation is there has to be a miscommunication somewhere because there "maximum entropy" isn't well justified by "Occam's razor".  In some sense they could be seen to be the opposite, as "Occam's razor" is often phrased as "preferring the simplest solution" and "maximum entropy" is the most-random solution (under some measure) and thus the least-compressible (under that measure), which could be seen as the "most complicated" solution!  Regardless, none of this is necessary, because any method of choosing between equilibria is fine for the purposes of this paper.

---

> ### Author Response · Authors · 2022-08-02
> **Reviewer #2 Rebuttal**
>
> Thank you for the feedback and constructive criticism, it has certainly strengthened our paper. We have done our best to answer your questions and have some clarifications. In particular, we would like to point out that we did perform transfer experiments (see below). We hope that you will have a more positive view of this paper after our response.
>
> ### Strengths and Weaknesses
>
> Clarification: Although the constraints are linear, the objective is not. So the problem we are solving is a convex nonlinear, linearly constrained optimization problem, not an LP (linear objective, linear constraints).
>
> ### Motivation
>
> PSRO is usually used to solve for normal-form solution concepts in extensive-form games (sometimes called Double Oracle when solving normal-form games), and uses a normal-form NE (or (C)CE in some variants) as its “meta-solver” step. Therefore PSRO requires an equilibrium solver to be prescribed as a subroutine. So PSRO itself does not solve normal-form (C)CEs - it just provides a mechanism to scale normal-form equilibrium solvers to extensive-form games.
>
> CFRM can be adapted to learn normal-form *time-average* (C)CEs. Therefore one would need to run many iterations, and take the average distribution to arrive at an equilibrium. Furthermore, we are not aware of any results which describe which equilibrium will be selected by CFRM, or whether that equilibrium is unique.
>
> The benefit of our approach is that it can (i) solve games very quickly in a single feed-forward neural network pass, and (ii) will solve uniquely for a pre-determined equilibrium (e.g. maximum entropy). This is not to mention other advantages such as that it is a differentiable model.
>
> There are many existing solvers available for (C)CEs. The motivation for our work is not that they do not exist, but that they are slow, which limits their use as primitives/inner loop/subroutines in MARL algorithms.
>
> ### Solving All Games
>
> Clarification: Payoff arrays are continuous, therefore, there are infinitely many games of a fixed shape. Therefore, we are not solving “all N by N games at once”. However, there is a learnable structure [Towards the PAC Learnability of Nash Equilibrium, Duan, Arxiv, 2022], so it is possible to train a model that is capable of approximating the equilibrium solution over the space of all games. So after paying an upfront cost to training the model, we are able to very quickly approximately solve any game of that shape.
>
> ### Game-Size Scaling
>
> You are correct. When training over the full game space, as described in the paper, this approach does not scale to large games. The reason for this is that the size of payoffs becomes prohibitively large, and therefore the activations of the networks become prohibitively large, and  memory quickly becomes a problem.
>
> There are two points we would like to make in response to this. Firstly we think the approach is still useful because it can solve small and medium games very quickly, in batches, to unique equilibrium selection targets. This makes it applicable to MARL algorithms, such as Correlated Q-Learning [Greenwald, 2003, ICML], which would have been difficult to scale before. In this algorithm it is necessary to calculate the policy by mapping a multiplayer Q-Table (a normal-form game)  to an equilibrium solution for every state in the game. This needs to be recalculated every time the Q-Table is updated, or when using function approximation every time you want to do an update or take an action. The Correlated Q-Learning paper considers 2 player games with 4 actions each. Using an off-the-shelf iterative solver to solve each Q-table may take ~0.02 seconds per Q-table, or 50 Q-tables per second. Our 4x4 network can solve 4,000,000 Q-Tables per second.
>
> Secondly, note that the scaling problems are purely caused by the size of the representation of games. We chose the ambitious task of solving over the space of all games of a specific shape, which necessarily means using a dense payoff representation, and therefore many activations in each layer. Meanwhile other aspects of the network have good scaling properties: (i) the output of our neural network, the dual variables, are very space efficient, (ii) the number of parameters in the network is independent of the number of actions in the game (see Section 4.4 - it only depends on the number of channels in each layer), and (iii) the equivariant architecture means that each sample is equivalent to training on all permutations of a game. Therefore to scale this work to larger games, one only needs to consider sparser representations of the game input. Polymatrix or graphical games could be a good place to start for future work.
>
> Update: We have included two new experiments to help mitigate the scaling concerns: (i) first is a pure scaling experiment, (ii) second is a generalization experiment showing zero-shot generalization to larger games. Please take a look under “Scaling” and “Generalization” experiment sections.

---

> > ### Author Response · Authors · 2022-08-02
> > **Reviewer #2 Rebuttal Continued**
> >
> > ### Evaluation Distribution
> >
> > You are correct that evaluating on the “all games” distribution could be problematic because (i) the space could contain many uninteresting games, and (ii) the space could contain many easy games.
> >
> > This is why we do evaluate our approach on more-specific classes of games (Figure 4). Specifically, we train our model on “all games” distribution, and then evaluate on other distributions taken from GAMUT. We never train directly on the GAMUT game distributions. Furthermore, these are not cherry-picked distributions, we use the full set of 22 used elsewhere in the literature.
> >
> > We believe that this is the hardest possible stress test for our model. Either the “all games” distribution is easy, in which case transfer would be difficult, or “all games” is sufficiently complex/representative in which case transfer should be straightforward.
> >
> > The empirical results show good transfer to the GAMUT  distributions, some with worse gaps than L2 disc, and some with better gaps than L2 disc. We have clarified this in Figure 4 as we are concerned that we were not explicit enough that this was a transfer experiment.
> >
> > We take your point that practitioners may want to focus on subclasses of games. It is absolutely possible to use our approach to only learn a subclass of games.
> >
> > ### Occam's Razor
> >
> > Our original intended meaning was that “simple” often means “makes fewest assumptions”. From a Bayesian perspective, in the absence of other information, a uniform (equiv. maximum entropy) prior is often chosen. However we agree that the section, as written, is unnecessarily hand-wavey. We have rephrased this section and removed the term “Occam's razor”. As you point out, it is not necessary for the point we are trying to make.
> >
> > ### Hyper-parameters
> >
> > Hyperparameters were selected over a coarse sweep (over learning rate, number of layers), for 4x4 games and selected to minimize the “Solver Gap” metric. Evaluation is in-distribution / infinite data regime, so no cross-validation is necessary. Batch size was chosen to maximize utilization of our hardware. We then used the same architecture/hyper-parameters for all the non-baseline experiments in the paper, so the hyper-parameters are not particularly optimized. The linear architecture baseline was chosen to have the same number of layers as the equivariant network, and a similar number of parameters. Baseline experiment hyper-parameters were selected over their own sweeps.

---

> > > ### Comment · Reviewer_duLd · 2022-08-03
> > > **Reviewer #2 Rebuttal Response (part 3/3)**
> > >
> > > > This is why we do evaluate our approach on more-specific classes of games (Figure 4). Specifically, we train our model on “all games” distribution, and then evaluate on other distributions taken from GAMUT. We never train directly on the GAMUT game distributions. Furthermore, these are not cherry-picked distributions, we use the full set of 22 used elsewhere in the literature.
> > >
> > > Sorry, I believe I wasn't clear/specific enough with my concern.  I was aware that the GAMUT set was a transfer distribution, my concern was that it is unclear how well suited they are to understanding the transfer behavior of this particular type of algorithm (since it is quite a departure from prior work).  My impression is that GAMUT was designed as an evaluation suite for algorithms which tend to not rely on neural nets and solve the games one-by-one.  It is initially unclear how this holds up as an evaluation suite for a pretrained-network based algorithm, which we should expect to have different edge cases.  Looking through the GAMUT set more in-depth now, I agree that it is adequate to show that the approach is achieving results which are at least useful in *some* interesting subset of the game space, so I will be increasing my score to weak accept.
> > >
> > > I believe that the transfer performance would be more clear if one or two of the games which are more tricky, or edge cases were explicitly explained, along with the solution found by the algorithm, as it it difficult to parse in aggregate and requires knowledge about the GAMUT games which are otherwise hard to interpret.  Another way of showing useful transfer performance is showing that it works with the sorts of games that we find in the course of other MARL algorithms (as you described in the other comment), which would also be powerful.

---

> > > > ### Author Response · Authors · 2022-08-03
> > > > **Reply to comment**
> > > >
> > > > > My impression is that GAMUT was designed as an evaluation suite for algorithms which tend to not rely on neural nets and solve the games one-by-one. It is initially unclear how this holds up as an evaluation suite for a pretrained-network based algorithm, which we should expect to have different edge cases.
> > > >
> > > > > I believe that the transfer performance would be more clear if one or two of the games which are more tricky, or edge cases were explicitly explained, along with the solution found by the algorithm, as it it difficult to parse in aggregate and requires knowledge about the GAMUT games which are otherwise hard to interpret.
> > > >
> > > > Ok, we now understand your point and it is a fair one. We will see if we can obtain MARL test-set games to evaluate on. Another way we could hunt for edge cases is to adversarially generate games with poor (C)CE gaps, as we can differentiate through the network.

---

> > ### Comment · Reviewer_duLd · 2022-08-03
> > **Reviewer #2 Rebuttal Response (part 1/3)**
> >
> > > Clarification: Although the constraints are linear, the objective is not. So the problem we are solving is a convex nonlinear, linearly constrained optimization problem, not an LP (linear objective, linear constraints).
> >
> > The author's are right here, this was a typo on my part.
> >
> > >PSRO is usually used to solve for normal-form solution concepts in extensive-form games (sometimes called Double Oracle when solving normal-form games), and uses a normal-form NE (or (C)CE in some variants) as its “meta-solver” step. Therefore PSRO requires an equilibrium solver to be prescribed as a subroutine. So PSRO itself does not solve normal-form (C)CEs - it just provides a mechanism to scale normal-form equilibrium solvers to extensive-form games.
> >
> > >CFRM can be adapted to learn normal-form time-average (C)CEs. Therefore one would need to run many iterations, and take the average distribution to arrive at an equilibrium. Furthermore, we are not aware of any results which describe which equilibrium will be selected by CFRM, or whether that equilibrium is unique.
> >
> > Both of these points are correct on the part of the authors.  The difference between extensive form and normal form games is relevant here because however a MARL method solves the normal-form sub-problems it can still run into bottlenecks when there are many actions at the single-step level.  In that framing, it would be more proper to compare against NFSP, but this method has even worse theoretical guarantees than CFRM, so the point of the authors stands.  There was originally a category error on my part.  ( I think this is a category-error that many readers may have, so it would probably be useful to preempt this by explicitly casting this as a useful subroutine for these other methods so the contrast is clear)
> >
> >
> > >The benefit of our approach is that it can (i) solve games very quickly in a single feed-forward neural network pass, and (ii) will solve uniquely for a pre-determined equilibrium (e.g. maximum entropy). This is not to mention other advantages such as that it is a differentiable model.
> >
> > >There are many existing solvers available for (C)CEs. The motivation for our work is not that they do not exist, but that they are slow, which limits their use as primitives/inner loop/subroutines in MARL algorithms.
> >
> > I now agree that this would be good if it works reliably enough to replace other solvers in the same category.  My remaining concern then is about reliability.  Given that we're putting a neural net somewhere it previously wouldn't have been, this is effectively trading pre-computation costs, and the costs of errors, for the efficiency of the network over other exact-solution methods.
> >
> > > Clarification: Payoff arrays are continuous, therefore, there are infinitely many games of a fixed shape. Therefore, we are not solving “all N by N games at once”
> >
> > I'm confused by this comment.  I understand that there are infinitely many games given that the payoff matrices are continuous, but the network takes in the payoff matrices and (if it works) should output solutions for each one.  So thinking of the weights as, in some sense, containing all of the information necessary to quickly solve an arbitrary NxN game seems like an accurate description of the method, which would be intuitively hard than making a network which does this same thing for only one particular payoff matrix.
> >
> > > However, there is a learnable structure [Towards the PAC Learnability of Nash Equilibrium, Duan, Arxiv, 2022], so it is possible to train a model that is capable of approximating the equilibrium solution over the space of all games.
> >
> > The result that concerns me about the tractability is that approximating Nash Equilibrium are PPAD complete[1].  Given this, (if we believe that PPAD is a difficult computational class), we should think that finding a neural network which would solve for the all of the approximate Nash equilibrium for all games of a certain size would require:
> > 1) an exponential amount of pre-processing to find the weights
> > 2) would require an exponentially growing depth of the network
> >
> > In either case, it brings concerns that, due to the difficulty of the problem, we will not spend the exponential compute and so there should be some games which are not well approximated by the network.  However, this is fine if the games are ones we tend not to run into in practice.
> >
> > I do not know how the assumptions in that PAC Learnability paper relate to this result, but my impression is that it must be that the assumptions either over the hypothesis space or the distributions of games eliminate this case, and it is unclear to me if such assumptions would apply to the distribution of games we would expect to see in real applications.
> >
> > [1] On the Complexity of Approximating a Nash Equilibrium CONSTANTINOS DASKALAKIS,ACM Transactions on Algorithms 2013

---

> > > ### Author Response · Authors · 2022-08-03
> > > **Reply to Response**
> > >
> > > >  ( I think this is a category-error that many readers may have, so it would probably be useful to preempt this by explicitly casting this as a useful subroutine for these other methods so the contrast is clear)
> > >
> > > This is a useful point. We will be sure to adjust the story to ensure that this approach is presented to fill a certain niche: inner-loops/modules of algorithms which require repeated solving normal-form games to approximately-correct tolerances. Our approach is not intended to be a drop in replacement for all NE/(C)CE solvers. Nevertheless, we believe that this is an important/exciting niche, and will unlock/scale many new approached in MARL.
> > >
> > > > I now agree that this would be good if it works reliably enough to replace other solvers in the same category. My remaining concern then is about reliability. Given that we're putting a neural net somewhere it previously wouldn't have been, this is effectively trading pre-computation costs, and the costs of errors, for the efficiency of the network over other exact-solution methods.
> > >
> > > You are correct to point out there is a trade-off here, the huge speed up is not coming for free. Please see our expanded comment on Correlated-Q for further motivation.
> > >
> > > > I'm confused by this comment. I understand that there are infinitely many games given that the payoff matrices are continuous, but the network takes in the payoff matrices and (if it works) should output solutions for each one. So thinking of the weights as, in some sense, containing all of the information necessary to quickly solve an arbitrary NxN game seems like an accurate description of the method, which would be intuitively hard than making a network which does this same thing for only one particular payoff matrix.
> > >
> > > Apologies, we think we misunderstood your original criticism. You are of course correct that training on a smaller space of games would be easier. As you understand, we could train on a subset of games, and this may often be desirable in practice. There is nothing in other approach that prevents this. We focussed on "all games" in this paper because the thought it was the most interesting/flexible setting.
> > >
> > > > In either case, it brings concerns that, due to the difficulty of the problem, we will not spend the exponential compute and so there should be some games which are not well approximated by the network. However, this is fine if the games are ones we tend not to run into in practice.
> > >
> > > We take your point that equilibria are known to be hard to compute and that computing many is necessarily even harder. However, there is one property that works in our favour: neural networks can generalize over unseen games and exploit structure in the payoffs to learn an efficient mapping. For example, there are many different ways a payoff could have a pure joint strategy that dominates all others in the game. For NEs and CEs, the NN would only need to learn a very simple rule to accurately predict equilibria in such games. We agree that there may be edge cases that the NN struggled to predict.

---

> > > > ### Comment · Reviewer_duLd · 2022-08-09
> > > > **Raising my score to weak accept**
> > > >
> > > > I have found myself repeatedly coming back to thinking about this paper over the past several days. I think the insight and direction here is fundamentally important, though it is initially difficult to see because it originally seems very niche and is far from traditional approaches.  While I do not know how well it scales or how it well it would work in the context of a larger MARL algorithm, the discussion has convinced me that it is a direction that ought to be explored by the broader community and that NES is a significant step doing so.
> > > >
> > > >  I think that the paper would be greatly improved by assuming no MARL background and walking readers through to why this would be useful for MARL applications as an inner-loop matrix-game solver, since these applications would require high throughput and only approximate accuracy.  I think this would be extremely useful even if your audience is only MARL readers.
> > > >   Since this is an often ignored aspect of MARL algorithms I didn't realize that this was the intervention that was being suggested, even though I spend most of my time thinking about MARL algorithms, and I don't think I would be alone in that respect.
> > > >
> > > > I have included below some of the sections of the discussion that were useful in helping me see the perspective of the paper:
> > > >
> > > > > This is a useful point. We will be sure to adjust the story to ensure that this approach is presented to fill a certain niche: inner-loops/modules of algorithms which require repeated solving normal-form games to approximately-correct tolerances. Our approach is not intended to be a drop in replacement for all NE/(C)CE solvers. Nevertheless, we believe that this is an important/exciting niche, and will unlock/scale many new approached in MARL.
> > > >
> > > > >> I am not an expert on MARL, but I can see that could be a compelling application of NES. Could you make this clearer in the introduction? (I see you have added a sentence on this to Section 7, but I think it would help readers appreciate the significance of your work if your also mentioned this at the beginning of the paper).
> > > >
> > > > >Thank you for the suggestion. We are happy to include this as a motivating example.
> > > >
> > > > >We actually developed this network specifically to train agents using Correlated Q-Learning....

---

> > ### Comment · Reviewer_duLd · 2022-08-03
> > **Reviewer #2 Rebuttal Response (part 2/3)**
> >
> > > Firstly we think the approach is still useful because it can solve small and medium games very quickly, in batches, to unique equilibrium selection targets. This makes it applicable to MARL algorithms, such as Correlated Q-Learning [Greenwald, 2003, ICML], which would have been difficult to scale before. In this algorithm it is necessary to calculate the policy by mapping a multiplayer Q-Table (a normal-form game) to an equilibrium solution for every state in the game. This needs to be recalculated every time the Q-Table is updated, or when using function approximation every time you want to do an update or take an action. The Correlated Q-Learning paper considers 2 player games with 4 actions each. Using an off-the-shelf iterative solver to solve each Q-table may take ~0.02 seconds per Q-table, or 50 Q-tables per second. Our 4x4 network can solve 4,000,000 Q-Tables per second.
> >
> > Is this experiment available somewhere or is this hypothetical?  I agree that this example makes the case for the speed of the algorithm in the context of a MARL training pipeline, but I'm unsure of the approximation errors of the neural network solver as opposed to the exact solver on that distribution of games would result in the same performance.  If it did not cause errors in the context of this algorithm then it would prove the utility of this method in practice.
> >
> > > Therefore to scale this work to larger games, one only needs to consider sparser representations of the game input. Polymatrix or graphical games could be a good place to start for future work.
> >
> > I agree this is a much stronger scaling argument, though it's hard for me to evaluate the success/utility of it without the corresponding experiments.

---

> > > ### Author Response · Authors · 2022-08-03
> > > **Reply to Response**
> > >
> > > Thank you for the quick response. We have included some further comments to hopefully alleviate your concerns on the solution-accuracy/execution-time trade-off.
> > >
> > > > Is this experiment available somewhere or is this hypothetical? I agree that this example makes the case for the speed of the algorithm in the context of a MARL training pipeline, but I'm unsure of the approximation errors of the neural network solver as opposed to the exact solver on that distribution of games would result in the same performance. If it did not cause errors in the context of this algorithm then it would prove the utility of this method in practice.
> > >
> > > We actually developed this network specifically to train agents using Correlated Q-Learning. Unfortunately we do not have a reference ready for the Q-Learning experiments yet. It is a large enough amount of work to have its own paper. However, we can give some intuition of what we have found so far. In terms of the approximation errors, we found these were not a big problem in practice. We believe this is because the Q-Tables are often approximated using a neural network function anyway (and even in the tabular setting, throughout the majority of training, the Q-tables are only noisy estimates of the true value function) - so investing orders of magnitude extra time into computing exact equilibria with an iterative solver on top of a partially learned / approximate normal-form game is not a good trade-off. Having a fairly good NES enables much faster training - meaning the RL/exploration can dominate training time again. Furthermore in a setting like this, the network need not be pre-trained. It can be trained alongside the Q-values because (i) having very accurate solutions is less important earlier in training, as the Q-values themselves are very inaccurate, and (ii) we can fully train or mix with the Q-Tables we are seeing in the environment we are learning in.
> > >
> > > We are aware that this paper under submission cannot be evaluated based on unpublished future research, but we include these notes to further motivate the use-case for this approach.

---

### Official Review · Reviewer_5ivh · 2022-07-12

**Rating:** 8
**Confidence:** 4
**Soundness:** 4 excellent
**Presentation:** 3 good
**Contribution:** 4 excellent

**Summary:**

The authors demonstrate a method to perform unsupervised amortized optimization of game solvers that seemingly leads to sharp speedups. They also design an architecture that respects the equivariances of the problem and perform ablations demonstrating that this is useful.

**Questions:**

- Your solver gap has not asymptoted on any of the graphs. Is there a small experiment you can run to check if they do asymptote in any reasonable number of iterations? (I am not asking you to run this experiment to improve review score) This would make clear if there’s significant progress to be made or if simply continuing to run your solver training would lead to continual improvement.
- Is lower-case w (used throughout section 4.4) defined anywhere in the text?
- Is there a section comparing the run-time of conventional solvers to your method? If not, this would be neat to include.
- What is “naive payoff sampling” in the ablations?

**Limitations:**

Yes.

**Strengths And Weaknesses:**

Strengths:
- The technique seems useful and I could imagine it being helpful in MARL settings
- That the objective is "unsupervised" (i.e. you don't need the actual optimal strategy) is very neat
- The invariant architectures should be useful for other attempts at this problem.

The main weaknesses are in the writing of the paper, I highlight some problems below:

- The superscripts in Line 133 come out of nowhere; for example, why does $\hat{\sigma}$ have the $L_1$ norm as a superscript? I’m not sure what these superscripts mean on a first read.
- Someone from an adjacent field might not know what an NxN game is, I think it’d be worth explaining.
- The citation links are broken and link back to the title instead of to the actual citation
- Are Table 1 and 2 actually included in this paper? I see them in the supplement, are those the same tables you’re referring to in the paper body?
- On figure 4 it’s not clear what “left” and “right” mean
- In Figure 4 caption it’s worth pointing out (as you do in the text) that the arms correspond to worst, mean, best
- The term MECCE is introduced but not defined anywhere in the main paper
- The stack function in Equation 12 does not appear to be defined in the main text
- The equivariant architectures section was quite hard to follow; it is unclear what the underlying logic behind each of the transformations is. I would love to see a more expanded version of this section in the appendix or with the extra page if the paper is accepted.
- Equivariant is misspelled on line 193
- Figure 4 is somewhat hard to read where the bars overlap, you might consider using different colors and making the arm sizes of one of the bars different than the other?

---

> ### Author Response · Authors · 2022-08-02
> **Reviewer #1 Rebuttal**
>
> Thank you for your review and your clarity/writing suggestions. We will continue to improve on writing quality for future versions of the paper.
>
> [Minor Comments]
> Thank you for the feedback. We have fixed or are working on these problems.
> We will work on the figures’ clarity for camera ready. We apologize for not getting around to it for rebuttal time.
> We are working on more Figures to aid the understanding of the equivariant architecture and plan to add it to the appendices.
>
> [Asymptote]
> We haven’t observed a clear asymptote yet. Part of the problem in hunting for one is that we are plotting on log-log graphs which can make small improvements over increasingly many iterations seem salient. We imagine that there is a ceiling to performance when network capacity is reached. We will attempt to run this experiment for camera ready time.
>
> [Lower-case w]
> They are the trained weights used in a linear transformation. We will add the definition to the text. Apologies for the omission.
>
> [Runtime]
> Figure 4 shows iterative solver runtime of about average 0.05s for a 8x8 game. Our network can solve a batch of 4096 games in 0.0025s.
>
> [Naive Payoff Sampling]
> Each element sampled from Uniform[-Z_m, +Z_m], where Z_m is a scale chosen to produce unit variance. We have clarified this in the paper.

---

> > ### Comment · Reviewer_5ivh · 2022-08-06
> > **Strong recommendation to add more writing to equivariant architectures section**
> >
> > Great, thank you for the response! I am still in favor of this paper being accepted but would like to strongly suggest that, if accepted, the authors make the equivariant architectures section of their paper more expansive / clearer as this was the most confusing part of the paper for this reviewer.

---

> > > ### Author Response · Authors · 2022-08-08
> > > **Reply**
> > >
> > > > Great, thank you for the response! I am still in favor of this paper being accepted but would like to strongly suggest that, if accepted, the authors make the equivariant architectures section of their paper more expansive / clearer as this was the most confusing part of the paper for this reviewer.
> > >
> > > We will absolutely improve this section. Thanks.

---

### Author Response · Authors · 2022-08-02
**To All Reviewers**

Thank you everyone for you comments, suggestions and constructive criticism. We have taken the feedback on board, have made, and will continue to make improvements. Please see the updated revision of the paper. We will respond to individual comments below.

---

### Meta-Review · Area_Chair_e6nJ · 2022-08-31

**Recommendation:** Accept
**Confidence:** Less certain

**Metareview:**

This paper introduces an Neural Network based Equilibrium Solver which utilizes a special equivariant neural network architecture
to approximately predict NEs, CEs, and CCEs of normal-form games. Experiments show the effectiveness of the proposed methods across multiple dataset. All reviewers support the acceptance of this paper.

While I agree on the merit of this paper worth acceptance, I'd also recommend authors to revise a bit in the final version regarding to the theoretical complexity of finding equilibrium. (1) in line 18 "solving for an equilibrium of a game can be computationally complex [9, 8]", in fact, the cited intractable results only apply to finding Nash in multiplayer general-sum games. Finding CE/CCE can be always done by LP, which is tractable, and can be guaranteed to finish in polynomial time; (2) this paper emphasize that prior methods may take an non-deterministic time to converge, while this method proposed in this paper gives determinism. However, it appears to be the methods proposed in this paper is not provided with guarantees to converge in certain time (thus without determinism either). It's better if the authors can clarify or modify corresponding arguments.



**Award:**

No

---

### Decision · Program_Chairs · 2022-09-14

Accept